# Oncogenic PKA signaling increases c-MYC protein expression through multiple targetable mechanisms

Gary KL Chan[1,2], Samantha Maisel[1,2], Yeonjoo C Hwang[1,2], Bryan C Pascual[1,2], Rebecca RB Wolber[1,2], Phuong Vu[3,4], Krushna C Patra[3,4], Mehdi Bouhaddou[5,6], Heidi L Kenerson[7], Huat C Lim[1,2], Donald Long[8], Raymond S Yeung[7], Praveen Sethupathy[8], Danielle L Swaney[5,6], Nevan J Krogan[5], Rigney E Turnham[1,2], Kimberly J Riehle[7], John D Scott[9], Nabeel Bardeesy[3,4], John D Gordan[1,2]*

[1]Division of Hematology/Oncology, Helen Diller Family Comprehensive Cancer Center, University of California, San Francisco, San Francisco, United States; [2]Quantitative Biosciences Institute (QBI), University of California San Francisco, San Francisco, United States; [3]Department of Medicine, Harvard Medical School, Boston, United States; [4]Massachusetts General Hospital Cancer Center, Boston, United States; [5]Department of Cellular and Molecular Pharmacology, University of California San Francisco, San Francisco, United States; [6]J. David Gladstone Institute, San Francisco, United States; [7]Department of Surgery and Northwest Liver Research Program, University of Washington, Seattle, United States; [8]Department of Biomedical Sciences, College of Veterinary Medicine, Cornell University, New York, United States; [9]Department of Pharmacology, University of Washington Medical Center, Seattle, United States

*For correspondence:
John.Gordan@ucsf.edu

Competing interest: The authors declare that no competing interests exist.

**Abstract** Genetic alterations that activate protein kinase A (PKA) are found in many tumor types. Yet, their downstream oncogenic signaling mechanisms are poorly understood. We used global phosphoproteomics and kinase activity profiling to map conserved signaling outputs driven by a range of genetic changes that activate PKA in human cancer. Two signaling networks were identified downstream of PKA: RAS/MAPK components and an Aurora Kinase A (AURKA)/glycogen synthase kinase (GSK3) sub-network with activity toward MYC oncoproteins. Findings were validated in two PKA-dependent cancer models: a novel, patient-derived fibrolamellar carcinoma (FLC) line that expresses a DNAJ-PKAc fusion and a PKA-addicted melanoma model with a mutant type I PKA regulatory subunit. We identify PKA signals that can influence both de novo translation and stability of the proto-oncogene c-MYC. However, the primary mechanism of PKA effects on MYC in our cell models was translation and could be blocked with the eIF4A inhibitor zotatifin. This compound dramatically reduced c-MYC expression and inhibited FLC cell line growth in vitro. Thus, targeting PKA effects on translation is a potential treatment strategy for FLC and other PKA-driven cancers.

## Editor's evaluation

The authors employed global kinome profiling to identify key effectors of protein kinase A (PKA) oncogenic signalling in fibrolamellar carcinoma and melanoma cell line models. Based on subsequent cell line-based validation using standard molecular and cellular biology assays, authors propose a model whereby the oncogenic effects of PKA are at least in part mediated by c-MYC. In addition to stabilizing c-MYC protein, the authors provide some evidence that PKA may stimulate c-MYC protein synthesis in an eukaryotic translation initiation factor 4F (eIF4F)-dependent manner.

Notwithstanding that the underlying mechanisms remain obscure, it was thought that this study is of broad interest inasmuch as it provides hitherto unacknowledged insights into the molecular underpinnings of oncogenic PKA signalling and accordingly, it was thought that this manuscript may be of interest to researchers in the fields of cancer research, therapeutics, signal transduction and molecular and cell biology.

## Introduction

Protein kinase A (PKA) is an evolutionarily conserved signaling enzyme with established roles in diverse physiological processes, including the regulation of growth, differentiation, and metabolism (*Turnham and Scott, 2016*). PKA is controlled by cyclic AMP (cAMP) generated by the activation of G protein-coupled receptor (GPCR) signaling. Genomic alterations in the components of the GPCR-PKA signaling pathway lead to constitutive activation of this kinase in many human diseases including cancer (*Taylor et al., 2013*), such as amplified ligands of upstream GPCRs (*Coles et al., 2020*; *McCudden et al., 2005*), point mutations in the G-protein subunit *GNAS* (*Patra et al., 2018*), inactivation of PKA regulatory protein PKA-RIα (*Yin et al., 2011*), and mutations that directly alter the activity of the PKA catalytic subunit (PKAc; *Berthon et al., 2015*). Elevated PKA activity as a consequence of *GNAS* or *PRKACA* mutations has been reported in a variety of endocrine tumors (*Salpea and Stratakis, 2014*). A prototypical example is the *PRKACA L205R* mutation, which generates an unregulated PKAc variant found in adrenocortical and ACTH(Adrenocorticotropic Hormone)-producing pituitary tumors in patients with Cushing's syndrome (*Cao et al., 2014*). Patients with germline inactivating mutations in *PRKAR1A* are predisposed to develop myxomas, thyroid, and gonadal tumors, referred to as Carney Complex (*Yin et al., 2011*). Recently, a *DNAJB1-PRKACA* gene fusion has emerged as the dominant oncogenic event in a rare liver cancer, fibrolamellar carcinoma (FLC; *Honeyman et al., 2014*). This genetic lesion is found in 79–100% of FLC (*Honeyman et al., 2014*; *Cornella et al., 2015*), with rare cases instead bearing *PRKAR1A* deletion (*Graham et al., 2018*). *DNAJB1-PRKACA* fusions have also been described in very small subsets of hepatocellular carcinoma (*Cancer Genome Atlas Research Network, 2017*), cholangiocarcinoma (*Nakamura et al., 2015*), and oncocytic biliary tumors (*Singhi et al., 2020*). Thus, oncogenic activation of PKA signaling is found in a substantial number of cancers.

The PKA holoenzyme is composed of two catalytic (C) and two regulatory (R) subunits (*Taylor et al., 2013*). In the inactive state, R subunits form a homodimer that binds and inhibits the C subunits. cAMP is generated by GPCR/Gαs-mediated stimulation of adenylyl cyclase. This diffusible second messenger binds R subunits, causing a conformational change that allows greater mobility and activity of the C subunits, while maintaining localization of active kinase complexes (*Smith et al., 2017*). The spatiotemporal specificity in cAMP signaling is provided by A-kinase anchoring proteins (AKAPs). This family of 60 human proteins sequester PKA at subcellular locations, creating nanodomains for relay and modulation of local cAMP signals (*Langeberg and Scott, 2015*; *Omar and Scott, 2020*).

PKA signaling modulates cancer-relevant processes including growth factor signaling, cell migration, cell cycle regulation, and control of cell metabolism. However, it remains unclear which oncogenic pathways downstream of PKA are essential and in which tumor types and contexts they have the greatest impact (*Burton and McKnight, 2007*; *London et al., 2020*). For example, DNAJ-PKAc stimulates ERK activation in an FLC model system (*Turnham et al., 2019*), operating via its interaction with AKAP-Lbc (*Smith et al., 2010*). In *GNAS*-mutant pancreatic tumor cells, PKA-mediated suppression of the salt-inducible kinases (SIK1-3) supports tumor growth (*Patra et al., 2018*). PKA has also been connected to control of the $G_2/M$ transition (*Grieco et al., 1996*; *Kotani et al., 1998*) and cell survival under glucose starvation (*Palorini et al., 2016*). Interestingly, PKA also has context-specific tumor suppressive functions including modulation of the Hedgehog and Hippo signaling pathways and is mutationally inactivated in a subset of cancers (*Iglesias-Bartolome et al., 2015*; *Tokita et al., 2019*).

Despite its oncogenic action in multiple tumor types, PKA is challenging to target directly with small molecules. The ubiquitous role of PKAc in normal physiology makes global inhibition of the kinase a challenge and selective inhibitors of this kinase have intolerable side effects (*Wang et al., 2022*; *Toyota et al., 2022*). This challenge is particularly unfortunate in the context of FLC, a disease of young adults with only limited reported impact of chemotherapy, immunotherapy, or targeted therapy to date (*Dinh et al., 2022*). Thus, a better understanding of the essential downstream PKA targets in individual tumor types is a more tractable path for therapeutic development. To gain insight

into oncogenic PKA signaling networks and identify potential drug targets, we have investigated effects downstream of PKA activation. Accordingly, we generated cell models with regulatable PKA activity and derived proteomic profiles of PKA signaling. We show that common downstream effects of PKA include increased c-MYC protein expression. In this report, we demonstrate that Aurora Kinase A (AURKA), glycogen synthase kinase (GSK)–3B and the eukaryotic Initiation Factor (eIF)–4B all link PKA and c-MYC. Of these, control of translation appears to exert the most important effect in FLC and is targetable with the clinical eukaryotic Initiation Factor 4A (eIF4A) inhibitor zotatifin, leading to reduced c-MYC protein expression and tumor cell viability.

## Results

### *PRKACA* alterations are common among tumor types

We first analyzed the frequency of PKA-activating somatic alterations in the TCGA Pan Cancer Atlas (*Weinstein et al., 2013*), including both *PRKACA* gain-of-function and *PRKAR1A* loss-of-function mutations in addition to copy number alterations across multiple cancers (*Figure 1A*). We found a frequency of *PRKACA* amplification of 0.3–11.3% and a rate of activating mutations of 0.2–2.7%. The greatest frequency of activation occurred in malignant peripheral nerve sheath tumors and ovarian cancers. *PRKAR1A* loss of function mutations were rarer, including both inactivating mutations (0.2–5.3%) and deep deletions (0.4–4%), that were predominantly detected in adrenocortical carcinoma (*Figure 1B*).

### Kinome profile of oncogenic PKA signaling

Cell lines with PKA-activating mutations were engineered for inducible PKAc activation or inhibition to study PKA signaling effects (*Figure 1C*). These models include the bladder cancer line 639V (*PRKACA* copy number gain *Barretina et al., 2012*) and Colo741 skin and ML1 thyroid (*PRKAR1A* frameshift mutations *Ghandi et al., 2019*) lines; of note, Colo741 was derived from a patient with colon cancer but is thought to be a melanoma (*Vincent and Postovit, 2017*). 639V and Colo741 have been profiled for PKA dependency in the Cancer Dependency Map program, with Colo741 highly dependent on *PRKACA*. Although not available when the proteomic analysis was performed, we also used FLX1, a novel cell line from a patient-derived xenograft FLC model (*Oikawa et al., 2015*). FLX1 contains a fusion of PKAc with biochemical gain-of-function and may have distinct signaling effects from other PKA-activating mutations (*Turnham et al., 2019*). To create stably inducible cell models for proteomic analysis, we introduced doxycycline (dox)-controlled *3xFLAG-PRKACA* or *PRKAR1A$^{G325D}$*, a dominant inhibitor of PKAc with impaired cAMP binding (*Viste et al., 2005*; *Willis et al., 2011*), into 639V, Colo741, and ML1 cells via lentiviral infection. Inducible expression of PKAc and PKA RIa was confirmed by immunoblot analysis using a phospho-PKA substrate antibody (*Figure 1D*).

The engineered cells described above were cultured with or without dox for 48 hr and analyzed with global phosphoproteomics and multiplex inhibitor bead (MIB) kinome profiing (*Coles et al., 2020*; *Donnella et al., 2018*; *Sos et al., 2014*; *Budzik et al., 2020*). Bioinformatic analysis was performed on the global phosphoproteomic data set with the Phosfate analysis tool to infer changes in kinase activity (*Ochoa et al., 2016*). These strategies allow us to measure known kinase/substrate relationships (Phosfate) and assay the activity of kinases whose substrates are not well known (MIBs). We initially confirmed the expected impact of *PRKACA* and *PRKAR1$^{G325D}$* constructs. Using the engineered 639V cell lines, we showed that phosphorylation levels of the PKA target VASP pS239 increased with *PRKACA* induction and decreased with *PRKAR1A$^{G325D}$* induction in our global phosphoproteomics analysis (*Figure 2A*). Similarly, we detected upregulation of PKAc with both Phosfate and MIBs platforms following dox treatment of *PRKACA*-inducible cells (*Figure 2B*).

We integrated the proteomics data in four categories: (1) Phosfate for cells with inducible *PRKACA* (*Figure 2C*, top), (2) Phosfate for cells with inducible *PRKAR1$^{G325D}$* (*Figure 2C*, bottom), (3) MIBs for cells with inducible *PRKACA* (*Figure 2D*, top), and (4) MIBs for cells with inducible *PRKAR1A$^{G325D}$* (*Figure 2D*, bottom). This analysis showed that YES, LYN, EPHB4, LIMK1, LIMK2, CDK5, and CDK7 kinase activities were reduced following PKAc overexpression and increased following PKA inhibition by *PRKAR1A$^{G325D}$* induction. We also saw that ROCK1 was upregulated by *PRKACA* and downregulated by *PRKAR1A$^{G325D}$* induction. These results provide proof of concept for our genetic model system. We focused on credentialed drug targets among the list of candidates: we observed upregulation of the

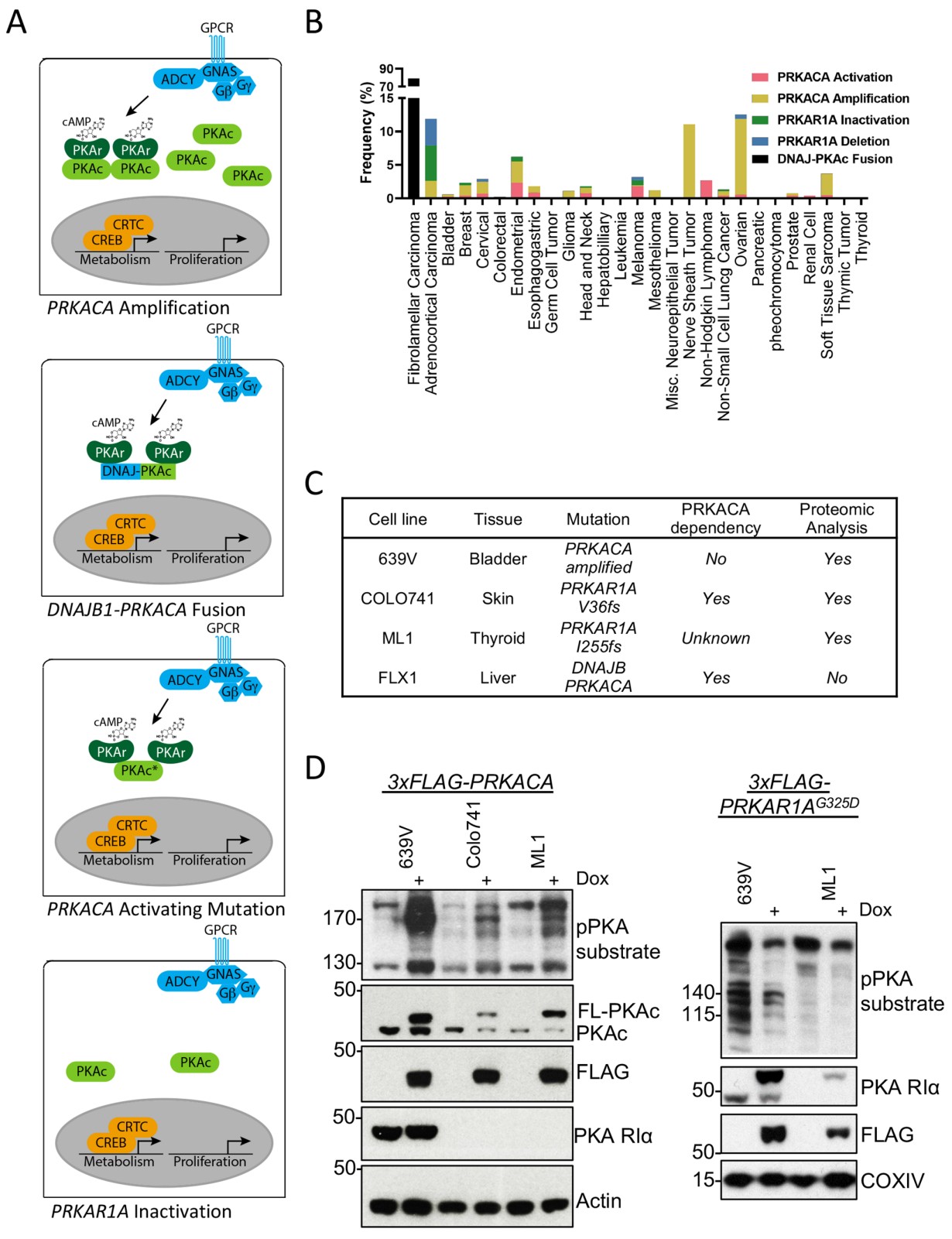

**Figure 1.** Recurrent PKA activating somatic alterations in human cancer. (**A**) Pathway illustrations of different PKA activating genomic alterations in order from top: *PRKACA* amplification, *DNAJB1-PRKACA* fusion, *PRKACA* activating mutation, and *PRKAR1A* inactivation or deletion. (**B**) TCGA PanCancer Project analysis showing the frequency of *PRKACA* gain-of-function (red and yellow) and *PRKAR1A* loss-of-function (green and blue) alterations in various cancer types. The reported frequency of *DNAJB1-PRKACA* fusion in fibrolamellar carcinoma (FLC) clinical samples is also included. (**C**) Cell lines

*Figure 1 continued on next page*

*Figure 1 continued*

used in this study, their PKA-related mutation, *PRKACA* dependency, and inclusion in proteomic analyses. (**D**) Immunoblots showing the change of PKA activity, as indicated by phospho-PKA substrate, in different cell lines with dox-inducible *3xFLAG-PRKACA* or *PRKAR1A^G325D^* with 1 μg/ml doxycycline (dox) for 48 hr. Left: engineered cell lines with inducible *3xFLAG-PRKACA*. Right: engineered cell lines with inducible *3xFLAG- PRKAR1A^G325D^*.

The online version of this article includes the following source data for figure 1:

**Source data 1.** Images for *Figure 1A*.

**Source data 2.** Table for *Figure 1B*.

**Source data 3.** Images for *Figure 1D* part 1/3.

**Source data 4.** Images for *Figure 1D* part 2/3.

**Source data 5.** Images for *Figure 1D* part 3/3.

pro-proliferative kinases AURKA, BRAF, and AKT2 by PKA (*Figure 2C*, yellow). Interestingly, the tumor suppressor STK11 was downregulated by PKA. Fewer signaling changes influencing proliferation were observed upon *PRKAR1A^G325D^* induction (*Figure 2C* bottom, 2D bottom).

Differences in isoform expression and shared kinase functions can obscure relationships between proteomic datasets. Thus, we used network propagation to integrate data across all of our cell models (*Cowen et al., 2017*), applying established pathway relationships from the ReactomeFI network to define connect activated kinases in the PKA-regulated kinome (*Gillespie et al., 2022*). Cytoscape was used to visualize PKA and its kinase network neighbors that are significantly altered by *PRKACA* or *PRKAR1A^G325D^* induction (*Figure 2E*), with kinases that are upregulated by PKA function marked as positive (red) and downregulated negative (blue). Non-kinase network nodes and non-PKAc-adjacent kinases were also found (*Supplementary file 3*). This analysis defined two PKA-dependent clusters with both networks including potential drug targets. One cluster is characterized by growth factor signaling effectors such as BRAF, multiple MAPKs, AKT, PKCs, and ERBB2. A second network emerged, with cell cycle kinases involved in the regulation of $G_2$/M including AURKA, PLK1, GSK3A/B, and several casein kinase family members. Importantly, both AURKA (*Walter et al., 2000*) and GSK3 (*Gregory et al., 2003*) have been previously described as PKA targets and can regulate MYC family proteins (*Dauch et al., 2016*; *Gustafson et al., 2014*; *Gregory et al., 2003*).

We confirmed key proteomic results by western blot in Colo741 and FLX1 cells treated with forskolin (FSK) and 3-isobutyl-1-methylxanthine (IBMX), to pharmacologically activate PKA. We observed strong activation of MAPK1/3 by FSK/IBMX in Colo741 and mild reduction in FLX1. Marked inhibition of GSK3B marked by phosphorylation of its inhibitory site serine 9 in FLX1, with a smaller, transient effect in Colo741 (*Figure 2F*).

## PKA signaling induces c-MYC and n-MYC expression in cell lines and tumor specimens

Our finding that PKA regulates AURKA and GSK3A/B suggested that MYC-family proteins might be responsive to PKA signaling. To extend these findings, we focused on our two key PKA-driven models, the FLX1 FLC line and Colo741, to determine whether PKA induces c-MYC or n-MYC expression. Cells were treated with FSK/IBMX for 0.5, 2, or 4 hr, leading to rapid phosphorylation of PKA substrates that correlated with progressive increase in c-MYC protein levels. Relatively low levels of n-MYC were detected in FLX1 but did increase as well (*Figure 3A*). Interestingly, sustained PKAc activation also resulted in mildly elevated *MYC* mRNA levels in FLX1 but not in Colo741 cells (*Figure 3B*).

Additionally, we generated an FLX1 cell line with dox-inducible 3xFLAG-*PRKAR1A^G325D^*, which produced the expected reductions in PKA substrate phosphorylation and c-MYC and n-MYC levels (*Figure 3C*). In control experiments, siRNA directed against PRKACA and dox induction of *3xFLAG-PRKAR1A^G325D^* greatly reduce the cell proliferation rate of FLX1 (*Figure 3—figure supplement 1A–B*). To confirm the relationship between PKA and MYC, we used the isogenic FLC model (*Turnham et al., 2019*), where an allele of the *Dnajb1-Prkaca* fusion was CRISPR engineered into AML12 murine hepatocytes. Immunoblot analysis confirmed that the engineered FLC clone had increased basal PKA activation, as well as higher c-MYC expression (*Figure 3D*). In an additional control, treating FLX1 with the PKA inhibiting tool compound H89 also reduced c-MYC levels (*Figure 3E*).

Finally, we assessed MYC protein levels in resected human FLC specimens. Immunoblot detection of the slower migrating DNAJ-PKAc fusion protein was used as a marker for FLC (*Figure 3E*, mid lower

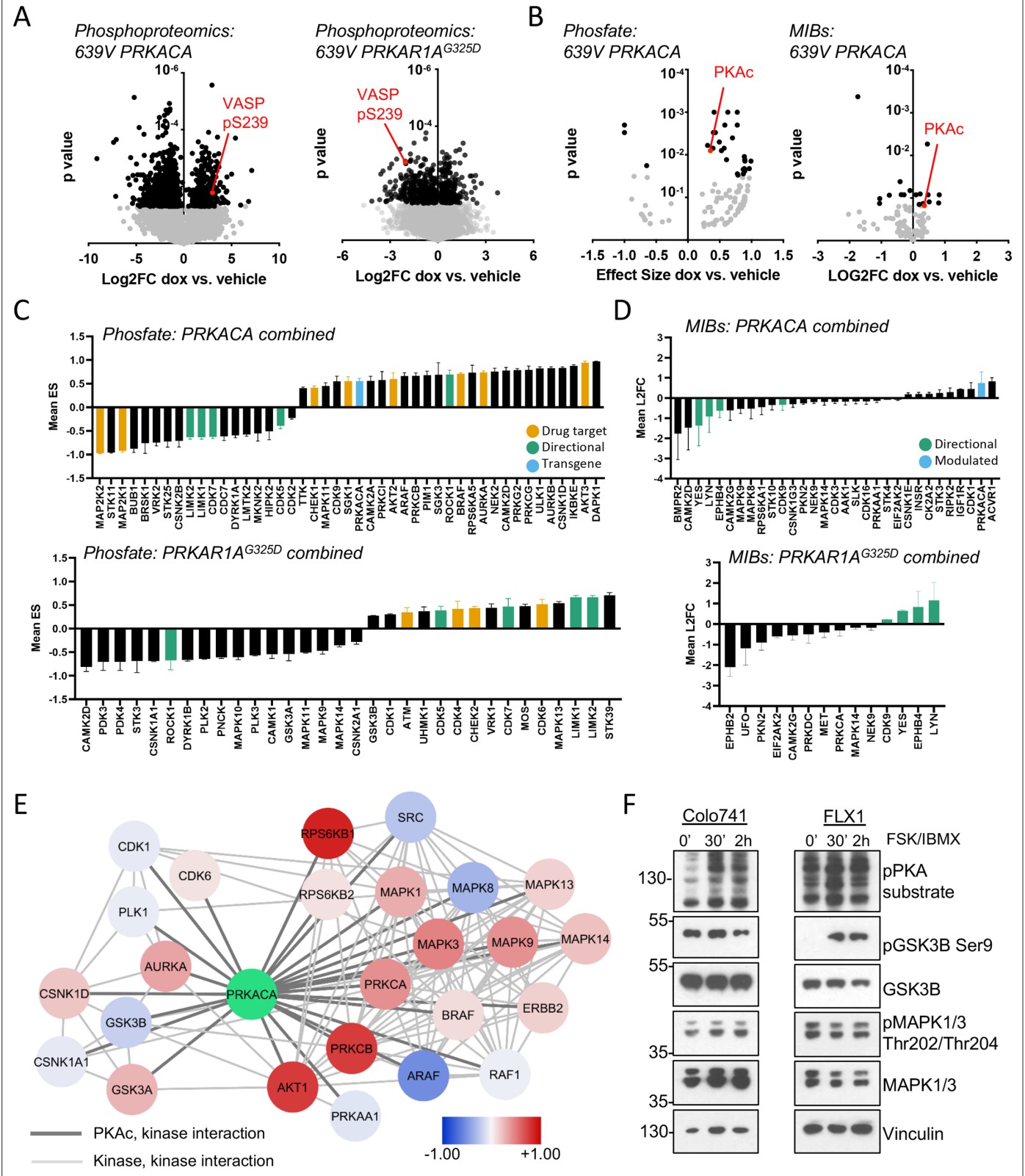

**Figure 2.** Kinome profiling to identify signaling nodes downstream of *PRKACA*. (**A**) Global phosphorylation changes in 639V with induction of *3xFLAG-PRKACA* or *3xFLAG-PRKAR1A^{G325D}*; VASP is shown as a positive control for PKA activation, technical replicates shown. (**B**) Change in kinase activity from Phosfate analysis or multiplex inhibitor beads (MIBs) pipeline using 639V with induction of *3xFLAG-PRKACA* compared to control; PKA catalytic (PKAc) shown as a positive control. Technical replicates shown. (**C**) Summary of overlapping activity in Phosfate data sets: effect size for all inferred kinases

*Figure 2 continued on next page*

*Figure 2 continued*

identified in at least two samples were averaged, shown with SD. Top panel shows results from *3xFLAG-PRKACA* induction, bottom panel from *3xFLAG-PRKAR1A^G325D^* induction. (**D**) Summary of overlapping activity in MIBs data sets: abundance of all bead-enriched kinases identified in at least two samples were averaged, shown with SD. Top panel shows results from *3xFLAG-PRKACA* induction, bottom panel from *3xFLAG- PRKAR1A^G325D^* induction. (**E**) Network integration of MIBs and Phosfate kinome profiles from 639V, Colo741, and ML1 with doxycycline (dox)-inducible *3xFLAG-PRKACA* and 639V and ML1 with dox-inducible *3xFLAG-PRKAR1A^G325D^*. Kinases marked in red show increased activity in *PRKACA* data sets and/or decreased activity in *PRKAR1A^G325D^* data sets, while those marked in blue show the converse. (**F**) Confirmation of PKA-induced signaling changes: Colo741 and FLX1 were treated with 50 μM forskolin (FSK)/3-isobutyl-1-methylxanthine (IBMX) for 30 or 120 min and then analyzed by immunoblot.

The online version of this article includes the following source data for figure 2:

**Source data 1.** Tables for *Figure 2A*.

**Source data 2.** Tables for *Figure 2B*.

**Source data 3.** Tables for *Figure 2C*.

**Source data 4.** Tables for *Figure 2D*.

**Source data 5.** Tables for *Figure 2E*.

**Source data 6.** Images for *Figure 2F*.

lane). Importantly, expression of this oncogenic PKAc form correlated with increased protein levels of both c-MYC and n-MYC (*Figure 3F*). To determine whether the relationship between PKA and MYC exists in additional cancers, we applied gene set enrichment analysis (GSEA) to RNASeq data from the TCGA adrenocortical carcinoma and serous ovarian carcinoma data sets. We compared all tumors with a genetic alteration conferring PKA activation to the tumors in the same dataset without genetic PKA activation. The higher rate of PKA-activating alleles in adrenal cancers allowed a more robust comparison, identifying multiple upregulated Hallmark Gene Sets, including MYC Targets V1 and V2. In the ovarian cancer dataset, MYC Targets V2 was in fact the only significantly upregulated gene set (*Figure 3G*). These data support a recurrent pattern of MYC activation by PKA.

## c-MYC effects on transcription and cell proliferation in PKA-driven cancers

To connect PKA- and MYC-driven gene expression effects on cellular behavior, we first performed RNASeq. This analysis compared a non-targeting control (NTC) siRNA to four pooled anti-*PRKACA* siRNA in FLX1 (*Figure 4A*, key targets highlighted). These caused a dramatic alteration in the FLX1 transcriptome, resulting in downregulation of Hallmark MYC Targets gene sets and upregulation of inflammatory and tumor suppressive gene sets (*Figure 4B*). Using individual siRNA, we knocked down *PRKACA* and *MYC*, confirming that both genes support the expression of the canonical c-MYC transcriptional target *ornithine decarboxylase* (*ODC*; *Figure 4C*). Because of its low level of expression in FLX1, *MYCN* knockdown is not shown. Control experiments did show a minor, inconsistent decrease in *MYC* mRNA levels following *PRKACA* knockdown (*Figure 4—figure supplement 1A*), which did not match effects on *ODC* and *cyclin D1* (*CCND1*). Similarly, treatment with FSK/IBMX caused a time-dependent increase in *ODC* and *CCND1* mRNA in FLX1 (*Figure 4D*).

We next tested the role of c-MYC in PKA-driven proliferation in Colo741 and FLX1 cells. Knockdown of *MYC* with four pooled siRNAs suppressed proliferation in FLX1 cells (*Figure 4E*). *MYC* siRNA knockdown also reduced proliferation in Colo741 cells, although to a lesser extent than in FLX1 (*Figure 4—figure supplement 1B*). Individual siRNAs were used to confirm the impact of *MYC* knockdown on FLX1 proliferation. AUC analysis of growth curves is shown, demonstrating that silencing *MYC* leads to a significant decrease in proliferation vs. NTC; *PRKACA* knockdown is shown for comparison (*Figure 4F*). Conversely, ectopic expression of *MYC* using a dox-inducible system increased proliferation of FLX1 cells (*Figure 4G*). Thus, c-MYC can play a significant role in the regulation of proliferation in PKA-dependent cancers.

## AURKA, PIM and GSK3B can influence c-MYC expression in PKA-driven cells

Our next objective was to dissect the signaling mechanisms that might control c-MYC protein expression downstream of PKA. To generate a dataset of broad utility, we first undertook a screen of 352 advanced kinase inhibitors to identify compounds that impact proliferation in FLX1 cells. We found

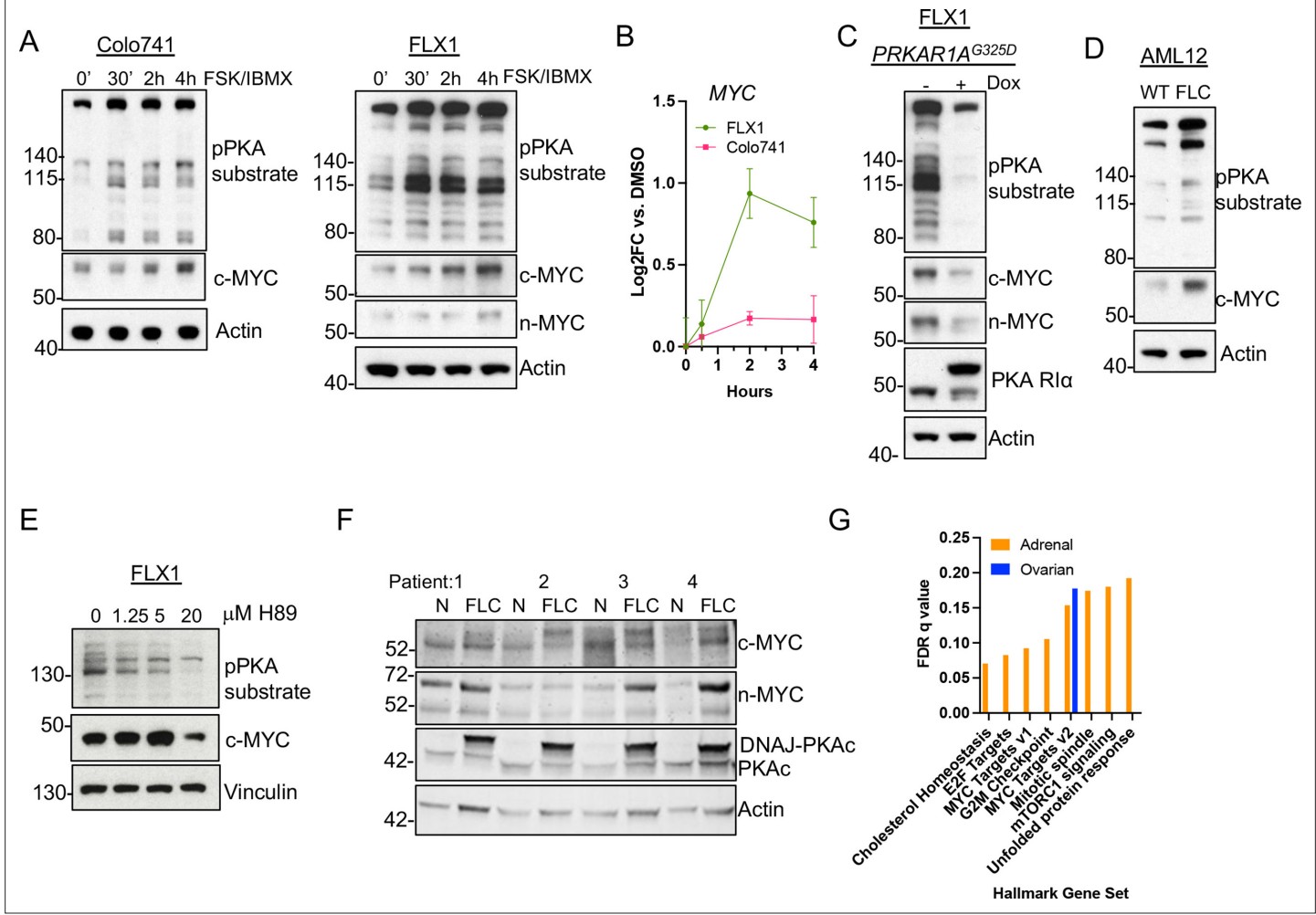

**Figure 3.** PKA activity correlates with c-MYC and n-MYC protein levels. (**A**) Immunoblots showing the change of PKA activity, as indicated by phospho-PKA substrate, and c-MYC and n-MYC expression in Colo741 and FLX1 cells after treatment with 50 µM forskolin (FSK) and 3-isobutyl-1-methylxanthine (IBMX) for 30 min, 2 hr, or 4 hr. n-MYC was not detected in Colo741 so is not shown. (**B**) Impact of 0–4 hr treatment with FSK/IBMX on *MYC* mRNA levels in Colo741 and FLX1; ± SD. Technical replicates from a representative experiment shown. (**C**) Immunoblots showing the change of PKA activity and c-MYC and n-MYC levels in engineered FLX1 cells with doxycycline (dox)-inducible *3xFLAG- PRKAR1A$^{G325D}$* with or without dox for 72 hr. (**D**) Immunoblots showing the basal level of PKA activity with phosphorylated PKA substrate and c-MYC expression in AML12 wild type (WT; left) and AML12$^{DNAJ-PKAc}$ cells (right). (**E**) Effect of 4 hr treatment with PKA-inhibiting tool compound H89 over a dose range from 1.25 to 20 µM on PKA substrate phosphorylation and c-MYC level. (**F**) Immunoblot showing the presence of DNAJ-PKAc and different level of c-MYC and n-MYC in fibrolamellar carcinoma (FLC) tumor samples (FLC) vs adjacent liver (**N**) from four FLC patients. (**G**) Summary gene set enrichment analysis (GSEA) of *PRKACA* amplified/mutant and *PRKAR1A* inactivated adrenocortical carcinoma or ovarian serous carcinoma vs. WT from TCGA. All significant 'Hallmark' gene sets are shown.

The online version of this article includes the following source data and figure supplement(s) for figure 3:

**Source data 1.** Images for *Figure 3A*.

**Source data 2.** Tables for *Figure 3B*.

**Source data 3.** Images for *Figure 3C*.

**Source data 4.** Images for *Figure 3D*.

**Source data 5.** Images for *Figure 3E*.

**Source data 6.** Images for *Figure 3F*.

**Source data 7.** Tables for *Figure 3G*.

**Figure supplement 1.** Effects of PKA inhibition on FLX1 cell proliferation.

**Figure supplement 1—source data 1.** Tables for *Figure 3—figure supplement 1A*.

**Figure supplement 1—source data 2.** Tables for *Figure 3—figure supplement 1B*.

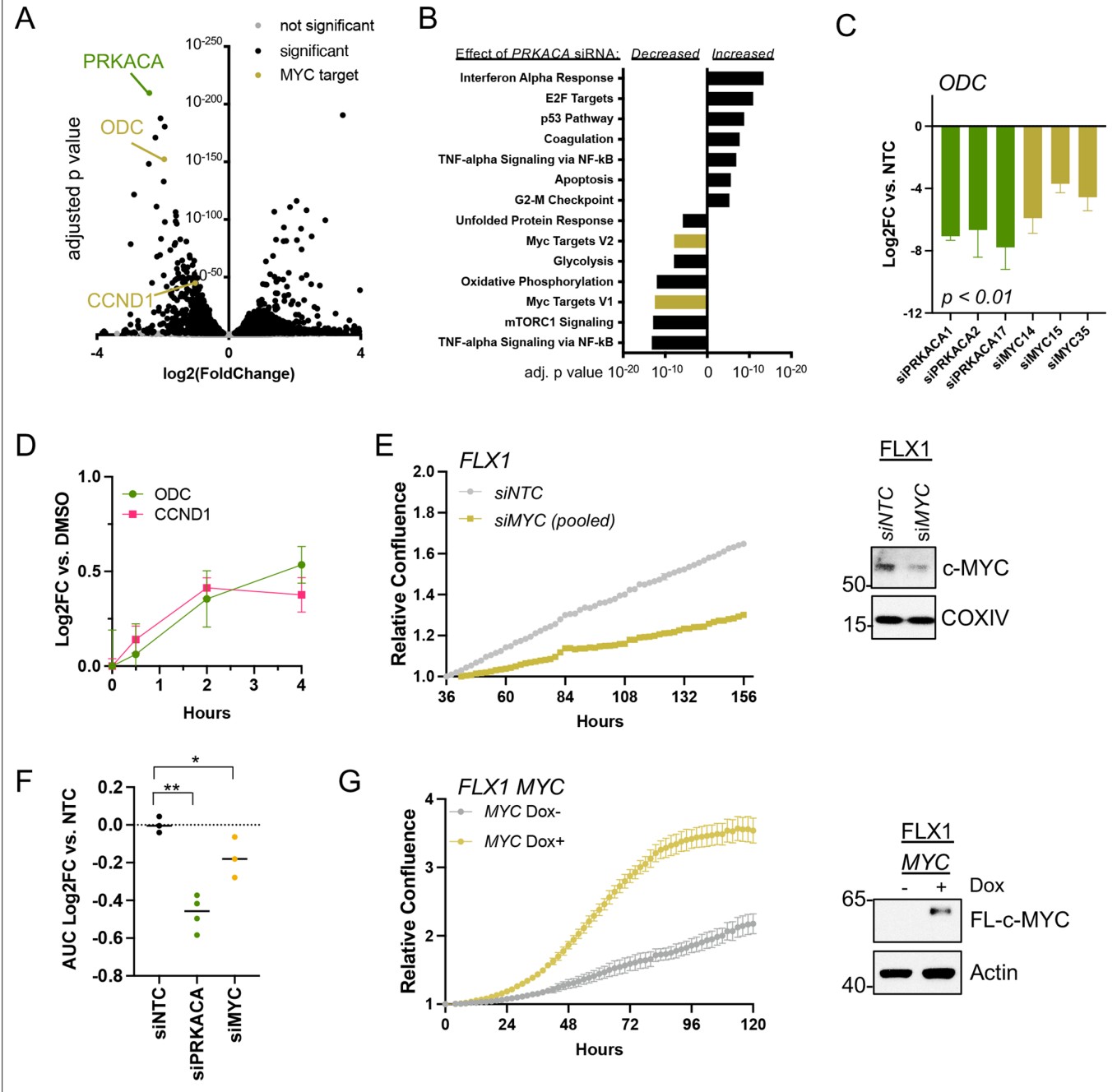

**Figure 4.** c-MYC alters transcription and proliferation in PKA-dependent cell models. (**A**) RNASEQ data from FLX1 cells after 48 hr treatment with four pooled siRNA against *PRKACA*. *PRKACA*, *ornithine decarboxylase (ODC)* and *Cyclin D1 (CCND1)* are highlighted. (**B**) Gene set enrichment analysis of Hallmark Gene Sets altered by *PRKACA* siRNA treatment. *MYC Targets V1* and *MYC Targets V2* are among the most significantly reduced. (**C**) Confirmation of overlapping effects of PKA and c-MYC on gene expression: FLX1 cells were transfected with individual siRNA directed against *PRKACA* and *MYC*. *ODC* expression was measured 48 hr later by quantitative RT-PCR (*MYC* and *PRKACA* knockdown shown in *Figure 1C*). Log(2)fold change vs. cells transfected with a non-targeting control (NTC) siRNA is shown ± SD; technical replicates shown from a representative experiment. p Value determined using two-tailed Student's t-test (**D**) Impact of 0–4 hr treatment with forskolin (FSK)/3-isobutyl-1-methylxanthine (IBMX) on *ODC* and *CCND1* mRNA levels in FLX1; ± SD. (**E**) Relative confluence of FLX1 cells in 96 well plates after *MYC* knockdown with four pooled siRNA. FLX1 cells were incubated 36 hr before recording to ensure attachment and then monitored with real-time microscopy for 120 hr. Experiments were done in duplicate, and representative results were shown with mean of technical replicates ± SD, n=10 for each condition; confirmation of knockdown shown by western blot. (**F**) Summary data of relative cell confluence shown as average AUC measurement from technical replicates of FLX1 treated with NTC or individual *PRKACA* and *MYC*-targeting siRNA. Representative results of technical replicates shown. p Value determined using two-tailed Student's t-test, *p<0.05 and ** p<0.001. (**G**) Relative confluence of engineered FLX1 cells with doxycycline (dox)-controlled *3xFLAG-MYC* in 96 well plates after treatment with

*Figure 4 continued on next page*

*Figure 4 continued*

or without 1 µg/ml dox. Experiment was duplicate, the representative results shown with mean of technical replicates ± SD, n=6 for each condition; induction confirmed by Immunoblot at 48 hr dox treatment. 3xFLAG-tagged c-MYC (FL-c-MYC) is shown.

The online version of this article includes the following source data and figure supplement(s) for figure 4:

**Source data 1.** Tables for *Figure 4A*.

**Source data 2.** Tables for *Figure 4B*.

**Source data 3.** Tables for *Figure 4C*.

**Source data 4.** Tables for *Figure 4D*.

**Source data 5.** Tables for *Figure 4E*.

**Source data 6.** Images for *Figure 4E*.

**Source data 7.** Tables for *Figure 4F*.

**Source data 8.** Tables for *Figure 4G*.

**Source data 9.** Images for *Figure 4G*.

**Figure supplement 1.** Data supporting c-MYC effects on transcription and proliferation.

**Figure supplement 1—source data 1.** Tables for *Figure 4—figure supplement 1A*.

**Figure supplement 1—source data 2.** Tables for *Figure 4—figure supplement 1B*.

**Figure supplement 1—source data 3.** Images for *Figure 4—figure supplement 1B*.

several Aurora kinase inhibitors in addition to the PKA-inhibiting tool compound H89 were particularly potent (*Figure 5A*). To illuminate potential PKA-regulated growth effects, we repeated this analysis in FLX1 cells upon induction of *PRKAR1A^{G325D}*. These experiments revealed that blocking PKA activity increased the potency of RTK, RAS/MAPK, and Aurora Kinase inhibitors, while PI 3-kinase/mTOR pathway inhibitor effects were diminished (*Figure 5B*). In addition, we identified three GSK3A/B inhibitors with differential activity following induction of *PRKAR1A^{G325D}*. Two compounds showed a minor increase in activity when PKAc was inhibited. The third, tideglusib, may have additional off-targets given its simple structure (*Mathuram et al., 2018*).

To expand this analysis beyond established drug targets, we next screened a kinome-wide siRNA library for modifiers of cellular proliferation in FLX1 cells. In a key control, the common essential genes WEE1 and PLK1 both showed a z-score of <–1. We identified a total of 30 kinases whose genetic depletion reduced cell proliferation with a z-score <–1 and 20 kinases that increased proliferation (*Supplementary file 7*). Sensitivity to *PIM2*, *EGFR*, *RPS6KB1*, and *PRKACA* knockdown were also noted, while *AURKA* knockdown did not significantly alter cell confluence. PIM2 is a serine/threonine kinase with similar substrates and function to AKT (*Fox et al., 2003*).

AURKA (*Dauch et al., 2016*; *Gustafson et al., 2014*), GSK3 (*Gregory et al., 2003*), and PIM2 (*Zhang et al., 2008*) are established regulators of MYC protein stability. To connect these screening results to c-MYC regulation, we first tested a panel of AURKA inhibitors against the PKA-dependent Colo741 and FLX1 cell lines. We noted that the conformation-disrupting AURKA inhibitor (CD-AURKAi) CD532 had the strongest effect on cell viability. This agent inhibits AURKA catalytic activity and also alters its conformation, resulting in destabilization of c-MYC and n-MYC (*Dauch et al., 2016*; *Gustafson et al., 2014*). Increasing our interest in this class of AURKA inhibitors, the partial CD-AURKAi MLN-8237 (alisertib) showed some effect in Colo-741, albeit not in FLX1 (*Figure 5D*, *Figure 5—figure supplement 1A*). Importantly, drug sensitivities (EC50=217.3 nM for Colo741; 692.8 nM for FLX1) matched the reported dose range for AURKA kinase inhibition (*Gustafson et al., 2014*). As a control, we confirmed that FSK/IBMX can increase c-MYC expression levels in an FLX1 cells treated with nocodazole. AURKA pT288 was increased by nocodazole and nocodazole +FSK/IBMX, but we did not detect total AURKA. AURKA pT288 did not correlate with increased c-MYC levels, and nocodazole did not block PKA effects on c-MYC (*Figure 5—figure supplement 1B*). We next tested a collection of PIM1/2 inhibitors on FLX1 viability (*Figure 5E*). We found that CX6258 and SGI1776 can each reduce c-MYC protein levels in FLX1, although this effect is overwhelmed by chronic stimulation of cAMP production (*Figure 5F*).

As inhibitors of both kinases only exerted partial effects on c-MYC levels, we tested combinations of PIM and AURKA inhibition. We first confirmed that both CD532 and MLN8237 alone can reduce c-MYC expression in FLX1 and Colo741. We note an off-target effect of CD532 on PKA activity, which

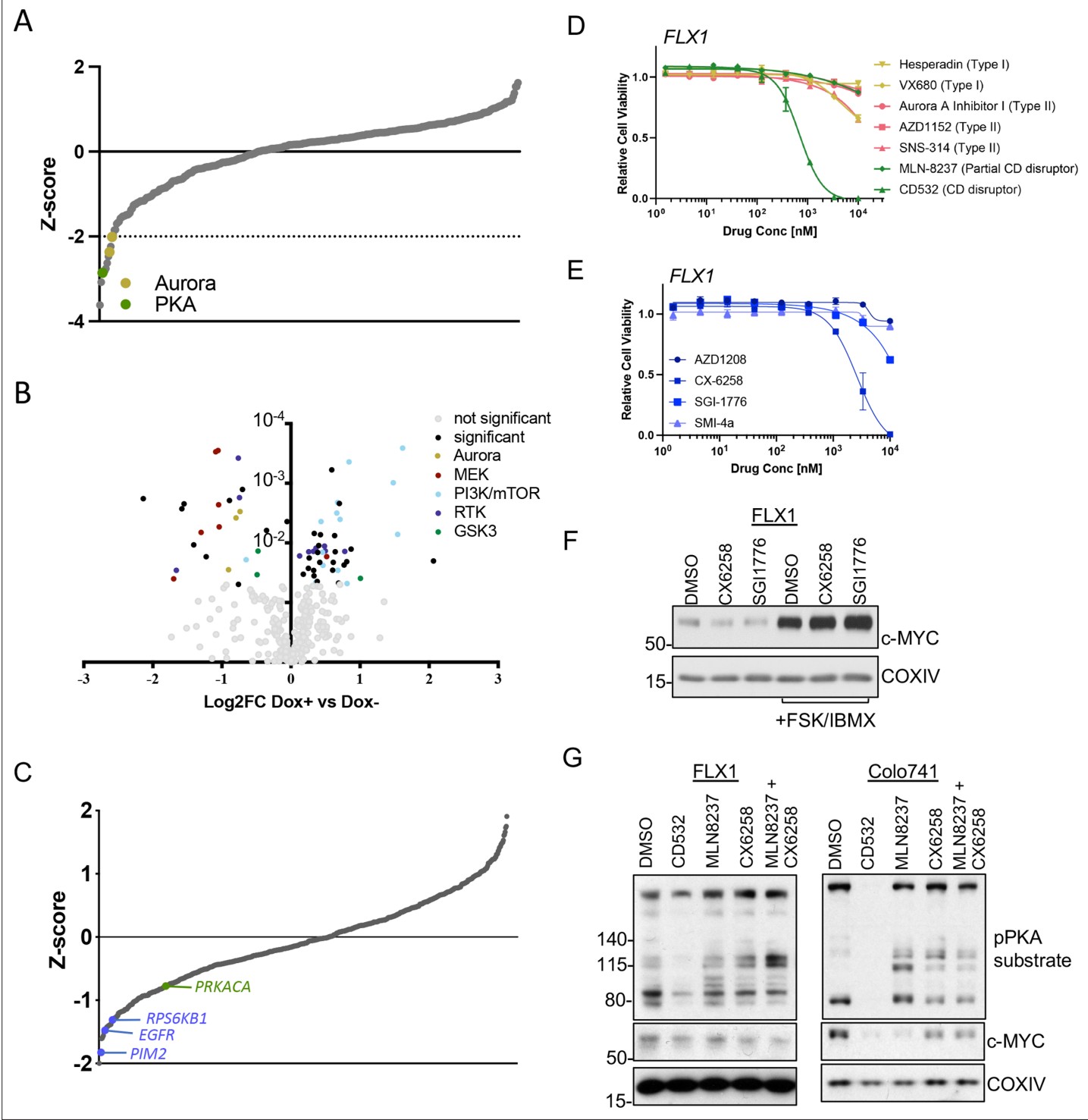

**Figure 5.** AURKA and GSK3B regulate c-MYC in PKA-dependent cell models. (**A**) Summary data from the FLX1 cell line treated with 352 kinase inhibitors from an advanced clinical compound library at 2 μM for 120 hr. The targets of selected compounds with a z-score ≥2 are highlighted, with the PKA inhibiting tool compound H89 shown. Average of three biological replicates is shown. (**B**) Impact of doxycycline (dox) induction of *3xFLAG-PRKAR1A^{G325D}* on drug sensitivity in FLX1: Cells were incubated with 1 μg/ml dox overnight and compound added on the following day; log2FC vs. median was derived ± dox, and then subtracted to identify those compounds whose activity was altered by *3xFLAG-PRKAR1A^{G325D}*. Data are averaged from three biological replicates. Inhibitors with p<0.05 were marked. Selected inhibitors were color coded based on their targets. (**C**) Kinase pooled siRNA library screen with FLX1 in 384 well plates shows the effect of each target kinase on cell proliferation (average of four biological replicates). Selected non-metabolic kinases that decrease cell proliferation with z-score –1 were marked. (**D**) FLX1 cells treated with dose curves of multiple AURKA

*Figure 5 continued on next page*

*Figure 5 continued*

inhibitors for 120 hr. Relative cell viability was measured by CTG assay vs. untreated control samples. Results are the mean ± SEM of triple biological replicates, three technical replicates per biological replicate. Inhibitors are color coded based on their binding mode. (E) FLX1 cells treated with dose curves of multiple PIM inhibitors as in B. (F) Effect of 24 hr treatment with 5 μM of different PIM inhibitors ±4 hr treatment with 50 μM forskolin (FSK)/3-isobutyl-1-methylxanthine (IBMX). (G) Immunoblot showing the change of PKA activity, as indicated by phospho-PKA substrate, and c-MYC and n-MYC levels in Colo741 and FLX1 cells after treatment with DMSO, 1 μM CD532, MLN8237, CX6258, or combination of 1 μM MLN8237 and 1 μM CX6258 for 24 hr.

The online version of this article includes the following source data and figure supplement(s) for figure 5:

**Source data 1.** Tables for *Figure 5A*.

**Source data 2.** Tables for *Figure 5B*.

**Source data 3.** Tables for *Figure 5C*.

**Source data 4.** Tables for *Figure 5D*.

**Source data 5.** Tables for *Figure 5E*.

**Source data 6.** Images for *Figure 5F*.

**Figure supplement 1.** Signaling effects on c-MYC in PKA-driven cells.

**Figure supplement 1—source data 1.** Tables for *Figure 5—figure supplement 1A*.

**Figure supplement 1—source data 2.** Images for *Figure 5—figure supplement 1B*.

**Figure supplement 1—source data 3.** Images for *Figure 5—figure supplement 1C*.

**Figure supplement 2.** Proteasome-independent PKA effects on c-MYC, (A) immunoblots showing the change of c-MYC protein in FLX1 cells after treatment with 50 μM forskolin (FSK)/3-isobutyl-1-methylxanthine (IBMX) and/or 20 μM MG132 for 2 hr.

**Figure supplement 2—source data 1.** Images for *Figure 5—figure supplement 2A*.

**Figure supplement 2—source data 2.** Images for *Figure 5—figure supplement 2B*.

**Figure supplement 2—source data 3.** Images and tables for *Figure 5—figure supplement 2C*.

may explain its potent effect on cell viability. Thus, we focused on MLN-8237 for combinations. Treatment with only MLN8237 was able to reduce c-MYC levels in Colo741, but not FLX1, and the PIM inhibitor CX6258 had only a mild cooperative effect with MLN8237 in reducing MYC levels in FLX1 cells (*Figure 5G*). Combination treatment with CX6258 and MLN8237 did not synergize to reduce viability in FLX1 (not shown).

Finally, as our data above (*Figure 3F*) show that GSK3B is phosphorylated on an inhibitory site by PKA and can regulate c-MYC degradation, we tested its impact on c-MYC levels. Pharmacologically blocking GSK activity with CHIR99021 significantly augmented the impact of PKAc activation with FSK/IBMX on c-MYC expression (*Figure 5—figure supplement 1C*).

Our finding show that numerous kinases converge on c-MYC protein stability but that single or combination inhibition fails to overwhelm PKA stimulation. Thus, we directly assessed the contribution of altered protein stability in PKA effects on c-MYC. Treating FLX1 cells with the proteasome inhibitor MG132 augmented the impact of FSK/IBMX on c-MYC levels (*Figure 5—figure supplement 2A*). Similarly, when PKA was inhibited with *PRKAR1A*^G325D induction, MG132 did not rescue c-MYC levels (*Figure 5—figure supplement 2B*). These results suggested that reduced degradation is not a major mechanism of PKA effects on c-MYC. Similarly, our proteomics did not reveal significant changes in c-MYC phosphorylation on T58, T62, or the putative PKA site S281 (*Padmanabhan et al., 2013*; *Supplementary file 2*). Thus, we tested c-MYC levels over time following treatment with cycloheximide (CHX) with or without FSK/IBMX treatment, finding no significant effect on c-MYC half-life following FSK/IBMX treatment (*Figure 5—figure supplement 2C*).

## PKA increases in c-MYC expression depend on eIF4A activity

These results raise the possibility that PKA could instead increase c-MYC translation. We performed GSEA on the altered phosphoproteins from our prior study of PKA signaling (*Coles et al., 2020*) and the phosphoproteomic data sets reported here. We observed statistically significant enrichment of proteins involved in translation initiation in all cases (*Figure 6A*). Our previous study showed that the eIF4F complex member eIF4B can be directly phosphorylated by PKA (*Coles et al., 2020*). Consistent with this, we observed increased eIF4B phospho-S422 by western blot following FSK/IBMX treatment

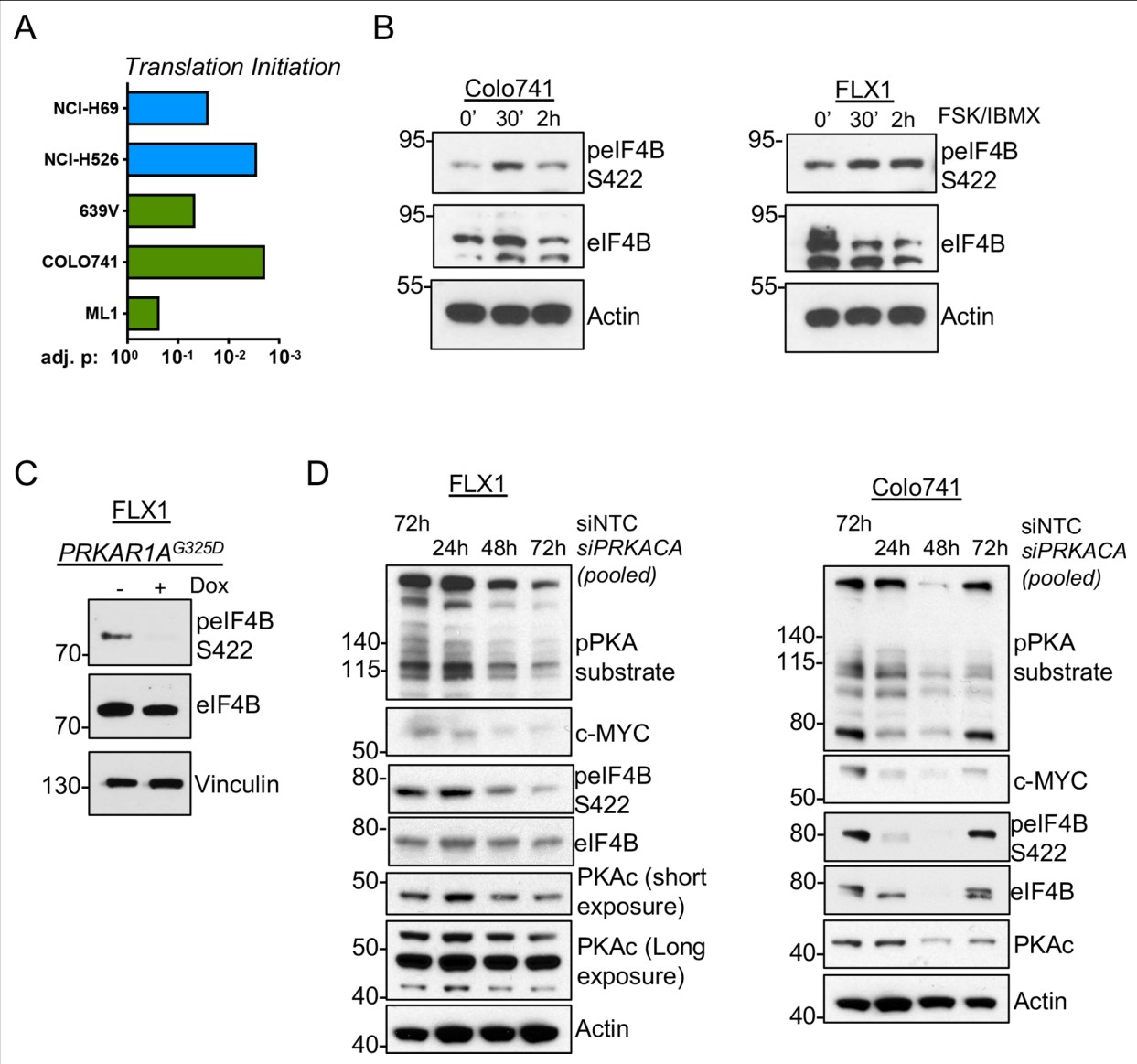

**Figure 6.** PKA signaling supports translation initiation. (**A**) Gene set enrichment analysis (GSEA) of significantly altered phosphoproteins following doxycycline (dox) induction of PKA in proteomics data from this study or chemical PKA stimulation in our prior publications. Results are shown for Hallmark gene sets, with statistically significant enrichment of altered phosphoproteins annotated to be involved in translation initiation. (**B**) Time course of forskolin (FSK)/3-isobutyl-1-methylxanthine (IBMX) in Colo741 and FLX1 showing increased phosphorylation of eIF4B Ser422. (**C**) Impact of 24 hr dox treatment on phosphorylation of eIF4B at Ser422 in engineered FLX1 cells with dox-inducible *3xFLAG-PRKAR1A^G325D*. (**D**) Immunoblots showing the change of PKA activity and PKAc and c-MYC expression and phosphorylation of eIF4B Ser422 in Colo741 and FLX1 cells after pooled *PRKACA* siRNA knockdown for 24, 48, and 72 hr vs. 72 hr with non-targeting control (NTC). Long and short exposures are used in FLX1 to show knockdown effect in wild type (WT) PKAc and the DNAJ-PKAc fusion.

The online version of this article includes the following source data for figure 6:

**Source data 1.** Tables for *Figure 6A*.

**Source data 2.** Images for *Figure 6B*.

**Source data 3.** Images for *Figure 6C*.

**Source data 4.** Images for *Figure 6D* part 1/4.

**Source data 5.** Images for *Figure 6D* part 2/4.

**Source data 6.** Images for *Figure 6D* part 3/4.

**Source data 7.** Images for *Figure 6D* part 4/4.

(*Figure 6B*). Conversely, eIF4B phosphorylation is reduced following induction of *PKAR1A*^G325D (*Figure 6C*) or pooled siRNA knockdown of *PRKACA* in either Colo741 or FLX1 cells (*Figure 6D*).

eIF4B phosphorylation at S422 increases the activity of the RNA helicase eIF4A (*Harms et al., 2014*), which unwinds the complex 5' untranslated regions (UTR) of multiple pro-growth genes including *MYC* (*Wolfe et al., 2014*). We found that the related eIF4A inhibitors rocaglamide and zotatifin markedly attenuate the induction of c-MYC by FSK/IBMX to near baseline levels (*Figure 7A*). Similarly, rocaglamide reduces the level of c-MYC to one similar to that achieved by induction of *PKAR1A*^G325D, with limited additive effect. Interestingly, napabucasin, which blocks eIF4E (*Zuo et al., 2018*) and has been described to reduce MYC levels in FLC (*Lalazar et al., 2021*), had relatively little effect in our system (*Figure 7B*). Furthermore, we found that protein levels of exogenously introduced c-MYC lacking a 5'UTR are not reduced by *PRKACA* knockdown with pooled siRNA (*Figure 7—figure supplement 1A*) or zotatifin treatment (*Figure 7C*; *Figure 7—figure supplement 1B*).

Finally, we assessed whether eIF4A inhibitor sensitivity was connected to a signaling effect of PKAc. Zotatifin potently reduced FLX1 and Colo741 viability with an EC$_{50}$ of 7 nM for FLX1 and 22 nM for Colo741. These concentrations are similar to those that predict in vivo potency for zotatifin in other cell lines (*Gerson-Gurwitz et al., 2021*). We further found that the impact of zotatifin is significantly blunted by *PRKAR1A*^G325D induction in FLX1 (*Figure 7D–E*), with siRNA knockdown of *PRKACA* and *MYC* also largely abrogating the effect of zotatifin on FLX1 proliferation (*Figure 7F*; *Figure 7—figure supplement 1C*). Zotatifin treatment also resulted in reduced expression of *CCND1* and *ODC*, mirroring the impact of *PRKACA* or *MYC* knockdown; *eIF4A2* is known to be induced by zotatifin (*Ho et al., 2021*) and is shown as a control (*Figure 7G*). Thus, PKA effects on c-MYC translation are amenable to therapeutic inhibition.

## Discussion

Over the last decade, tumor sequencing and mouse modeling studies have demonstrated the importance of GNAS/PKA signaling in cancer, including frequent oncogenic mutations in *GNAS* (*O'Hayre et al., 2013*) across multiple tumor types. Related studies have delineated the essential role of PKA as its effector (*Coles et al., 2020*; *Patra et al., 2018*). Here, we define the tissue distribution of genetic alterations in *PRKACA* and *PRKAR1A* that result in PKA activation in cancer and map the multiple conserved pathways downstream of oncogenic PKA signaling, many of which impinge on the expression of c-MYC (*Figure 7H*).

Our proteomic analysis has uncovered both expected and novel effects of PKA in cancer cell lines. We note that these findings may represent both direct and indirect effects of PKA, with PKA effects on the cell cycle and cellular metabolism potentially influencing kinase signaling due to changes in cell state. The analysis of kinase signaling recapitulated findings from previous studies, most notably activation of the AKT and RAS/MAPK pathways (*Coles et al., 2020*; *Turnham et al., 2019*; *Isobe et al., 2017*; *Dinh et al., 2020*) and inhibitory effects of PKA on various kinases, including STK11 (*Collins et al., 2000*) and its effectors. We also noted substantial effects on kinases involved in cell migration (e.g. YES, EPHB4, LIMK1, LIMK2, and ROCK1), with the majority being inhibited by PKA. These interesting observations merit further investigation for their mechanistic impact in PKA-associated malignancies. When data were integrated using network propagation, we found two key clusters in PKA-driven signaling, one driven primarily by the RAS/MAPK pathway and the other made up of multiple kinases involved in the G2/M transition, also influencing the stability of MYC-family proteins. These findings are supported by other studies demonstrating PKA effects on GSK3A/B (*Fang et al., 2000*) and upregulation of AURKA in GNAS (*Coles et al., 2020*) and DNAJ-PKAc-driven malignancies (*Simon et al., 2015*). Analysis of global phosphoproteomic data further revealed an activity of PKA toward mRNA translation, also seen in the upregulation of mTORC1 targets in our transcriptional analysis (*Figures 3G and 4B*). We found few reports connecting PKA to translation in mammalian cells other than our own previous work showing direct phosphorylation of eIF4B by PKAc (*Coles et al., 2020*), but multiple studies in yeast have demonstrated PKA effects on translation (*Leipheimer et al., 2019*), with reports from both yeast and plants that PKA impacts eIF4A (*Bush et al., 2016*).

A major objective of this study was to identify targetable signaling mechanisms downstream of PKA in FLC. This is particularly critical given that directly targeting PKA appears unlikely to be clinically possible due to the critical physiological functions of PKA. We found that upregulation of c-MYC, and to a lesser extent n-MYC, is an effect of PKA signaling. siRNA-mediated *MYC* knockdown decreased

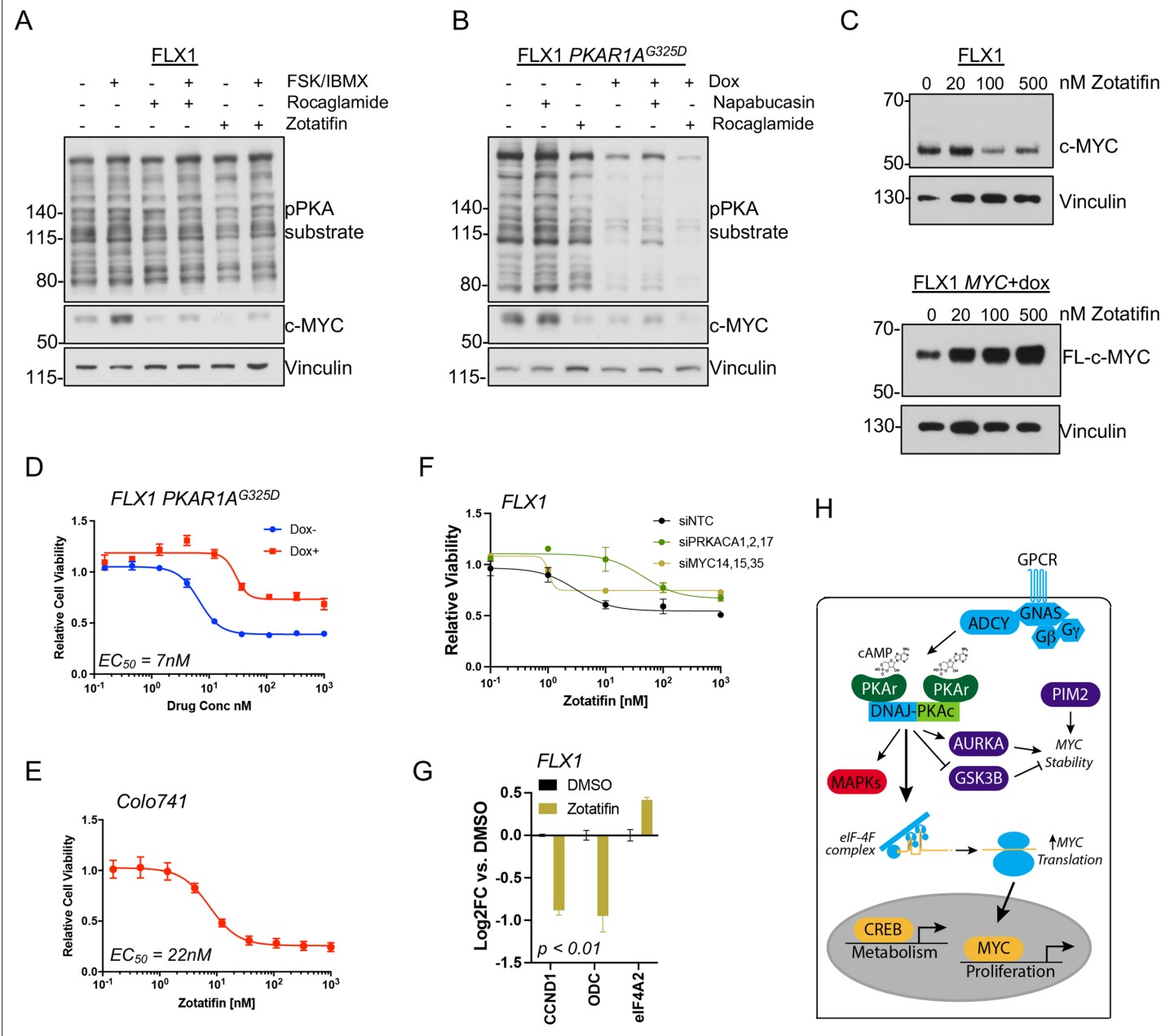

**Figure 7.** PKA effects on c-MYC are blocked by eIF4A inhibition. (**A**) Immunoblots showing the c-MYC protein levels in FLX1 cells after treatment with 100 nM rocaglamide or zotatifin for 24 hr and/or 50 µM forskolin (FSK)/3-isobutyl-1-methylxanthine (IBMX) for 4 hr. (**B**) Immunoblots showing the change of c-MYC level in engineered FLX1 cells with doxycycline (dox)-inducible *3xFLAG-PKAR1A^{G325D}* after dox induction for 48 hr and treatment with 1 µM napabucasin or 100 nM rocaglamide for 24 hr. (**C**) Parental FLX1 cells or FLX1 with 48 hr dox-induced *3xFLAG-MYC* lacking a 5'UTR treated with escalating doses of zotatifin for 24 hr and blotted for c-MYC. (**D**) Impact of 72 hr of a dose curve of zotatifin alone or in combination with dox-induced *3xFLAG- PKAR1A^{G325D}* on FLX1 viability by Cell Titer-Glo. Results from technical replicates of a representative experiment ± SD relative to DMSO for each curve. (**E**) Parental Colo741 treated as in D. (**F**) Zotatifin sensitivity tested in FLX1 as in D, following siRNA knockdown of PRKACA or MYC. Results combined from three siRNA. Quantitative RT-PCR (qRT-PCR) confirmation of knockdown is shown in *Figure 1C*. (**G**) Impact of 24 hr 100 nM zotatifin on c-MYC targets by qRT-PCR; eIF-4A2 induction is a known effect of zotatifin and is shown as a control. Mean of technical replicates from one representative experiment is shown ± SD, p value determined with two-tailed Student's t-test. (**H**) Schematic of DNAJ-PKAc mediating cell proliferation in fibrolamellar carcinoma (FLC) by increasing c-MYC expression by increased translation, with additional effects via GSK3B, AURKA, and PIM2.

The online version of this article includes the following source data and figure supplement(s) for figure 7:

**Source data 1.** Images for *Figure 7A*.

**Source data 2.** Images for *Figure 7B*.

*Figure 7 continued on next page*

*Figure 7 continued*

**Source data 3.** Images for *Figure 7C*.

**Source data 4.** Tables for *Figure 7D*.

**Source data 5.** Tables for *Figure 7E*.

**Source data 6.** Tables for *Figure 7F*.

**Source data 7.** Tables for *Figure 7G*.

**Source data 8.** Images for *Figure 7H*.

**Figure supplement 1.** Data supporting reversal of PKA effects on c-MYC by eIF4A inhibition.

**Figure supplement 1—source data 1.** Images for *Figure 7—figure supplement 1A*.

**Figure supplement 1—source data 2.** Images for *Figure 7—figure supplement 1B*.

**Figure supplement 1—source data 3.** Tables for *Figure 7—figure supplement 1A*.

proliferation in FLC, and to a small extent in the melanoma line Colo741. Of note, while FLCs rarely harbors additional oncogenic mutations (*Cornella et al., 2015*), Colo741 has an activating *BRAF* mutation (*Ghandi et al., 2019*) which may maintain its proliferation even when c-MYC expression is blocked. The overlapping results from these genetically distinct cell lines suggest that PKA specifically exerts an influence on c-MYC in carcinogenesis. That premise is supported by our finding of upregulated MYC target gene expression in PKA-activated adrenal and ovarian cancers in the TCGA. When transcriptional activation of c-MYC drives oncogenesis, it is often considered to be 'undruggable' (*Dang et al., 2017*). We hypothesized, however, when that c-MYC is induced by an oncogenic kinase that disrupting the upstream signaling to c-MYC could in turn block its effects.

Our small molecule and siRNA kinome screens identified AURKA, GSK3A/B, and PIM1/2 as potential regulators of c-MYC levels, given prior publications connecting them to MYC stability. We present data that PKA stimulation can result in inhibition of GSK3B, and others have shown PKA phosphorylation of AURKA (*Walter et al., 2000*), although effects in FLX1 are not fully clear. We did not identify an effect of PKA on PIM kinases. While inhibition of AURKA and PIM kinases somewhat reduced c-MYC in our cell models, they had minor effects on cell proliferation and were variable between the two lines tested. Similarly, no significant phosphorylation changes were seen in sites that regulate MYC degradation by proteomics, and proteasome inhibition did not abrogate the effects of PKA on c-MYC in FLC. Finally, PKA could increase c-MYC levels without altering its stability.

These observations, coupled with findings that PKA stimulation increases eIF4B phosphorylation, suggested that PKA effects on translation initiation might be responsible for its induction of c-MYC expression. Consistent with this, inhibition of eIF4A with the natural product rocaglamide, or its clinically used derivative zotatifin, significantly reduced c-MYC protein levels and potently inhibited proliferation of our cell models. These effects were confirmed to be at least partially dependent on PKA and c-MYC expression.

Our study has several key limitations. While a valuable feature of our genetic approach to modulate PKA signaling is the lack of off-target effects seen with commonly used PKA-modulating tool compounds, PKA signaling is well known to be precisely spatiotemporally regulated (*Bauman et al., 2006*; *Coghlan et al., 1995*) and our overexpression systems do not allow compartmentalized control of PKA signaling. We also note that while our FLX1 cells provide unique insight into the biology of FLC, they have significant limitations. Their long doubling time posed significant challenges in generating stably engineered cell lines, particularly with knockdown of growth mechanisms. Similarly, the FLX1 has proven more resistant to siRNA than our other exemplar line, Colo741 (*Figure 6D*), particularly when treated with individual rather than pooled siRNAs. Furthermore, our FLC clinical samples and the FLX1 cell line expressed both c-MYC and n-MYC. However, expression of n-MYC was quite low in FLX1, and it was not possible to clearly assess its regulation and contribution to FLC growth. Thus, the development of more precisely engineered PKA-driven cancer cell models is essential to provide genetic validation for the mechanisms that we have outlined using signaling and small molecule inhibitors, with more FLC cell models specifically also needed to confirm dependency on MYC proteins for proliferation. Such models would also enable detailed characterization of the biochemical methods by which PKA can influence mRNA translation.

This manuscript reports a network map of signaling downstream of oncogenic PKA. We use functional studies to prioritize signaling mediators for their effect on cell growth in PKA-driven cancers, with a focus on FLC models. While our focus in this study has been on FLC, the systems-level mapping of PKA effects in cancer may have distinct implications for other tumor types. This may include a more significant role for PKA effects on c-MYC stability, including via AURKA and PIM2. Whereas FLC has few secondary mutations, the common co-occurrence of PKA-activating mutations with those impacting RAS/MAPK signaling suggests that PKA effects on other targets such as the SIK kinases (*Patra et al., 2018*) may also have a more important role in other cancers. Similarly, PKA activation in *APC*-mutant colorectal cancer could exert important effects on CTNNB1 via inhibition of GSK3. Finally, given PKA's role in metabolism, its analysis in patient-derived tissues may yield additional nuance. In FLC, our results identify zotatifin as a potential mechanism-driven therapy for FLC and other PKA-driven cancers but require in vivo validation in multiple models to confirm their relevance. With more study, it may be possible to provide proof of concept that targeting MYC by inhibiting its translation is a potential treatment for patients with FLC or other PKA-driven cancers, for whom few options currently exist.

## Materials and methods

### Cell culture reagents and treatment

Human bladder 639V cells (DSMZ #ACC 413), human skin Colo741 cells (ECACC 93052621), and human thyroid ML1 (DSMZ #ACC 464) cells were maintained in Dulbecco's modified Eagle's medium (DMEM) supplemented with 10% fetal bovine serum (FBS), penicillin (100 U/ml), and streptomycin (100 U/ml). The murine hepatocyte AML12 wild type (WT) and AML12$^{DNAJ-PKAc}$ cell lines were developed as described previously by the Scott lab (*Turnham et al., 2019*). These cells were maintained in 50:50 DMEM/Nutrient Mixture F-12 (F-12) supplemented with 10% FBS, 0.1× ITS liquid media supplement, dexamethasone (0.1 μM), and gentamicin (50 μg/ml). FLX1 cells were derived in the Bardeesy lab from a human FLC tumor and xenografted to mice through dispersal and direct plating onto cell culture and maintained in RPMI with 50 ng/ml HGF(hepatocyte growth factor), 10% FBS, penicillin (100 U/ml), and streptomycin (100 U/ml). All cells were cultured in a 37°C incubator with 5% CO$_2$. Cells were tested for mycoplasma contamination routinely. Recombinant HGF was obtained from PeproTech; dexamethasone, FSK, gentamicin, IBMX, and 100× ITS liquid media supplement from Millipore Sigma; CD532, DMEM, DMEM/F-12, RPMI, FBS, Lipofectamine RNAiMAX Reagent, Opti-MEM, and penicillin-streptomycin from Thermo Fisher Scientific; Zotatifin from MedChemExpress; Rocaglamide, MLN8237, CX-6258, and kinase inhibitor library (L1200) from Selleckchem. The human protein kinase siGENOME siRNA library was obtained from GE Dharmacon. FuGENE 6 transfection reagent and CellTiter-Glo assay system were obtained from Promega. siGENOME single and SMARTpool siRNA targeting NTC, *MYC*, and *PRKACA* were purchased from Dharmacon.

For individual experiments, cells were seeded at 200,000 cells in 6 cm dishes overnight before treatment, except FLX1 cells, which grew for two nights. For drug treatment, a final concentration of 50 μM IBMX, 50 μM FSK, and 1 μM of the indicated drug were added to the cells in this order for the desired time periods and harvested, with the exception of zotatifin and rocaglamide, which were used at several doses. For siRNA treatment, 12 μl of 20 μM siRNA was added to the cells with Lipofectamine RNAiMAX reagent in Opti-MEM, incubated for 72 hr, and harvested. For CHX or MG132, a final concentration of 10 μg/μl and 20 μM, respectively, was added for the indicated time.

### DNA transfections and lentivirus production

Plasmids containing *PRKACA*, *PRKAR1A*, and *MYC* were obtained from the Human ORFeome v8.1 Collection (courtesy of Sourav Bandyopadhyay, UCSF or DNASU) and cloned into a gateway compatible version of pLVX-Tet-One (puro) with 3xFLAG tag at the N terminal (for *PRKAR1A* and *MYC*) or C terminal (for *PRKACA*). The *PRKAR1A*$^{G325D}$ single point mutation was introduced using standard PCR site-directed mutagenesis. The final plasmids were packaged in HEK 293T cells for 72 hr to produce lentivirus, which were used to establish cell lines with each respective transgene.

## SDS-PAGE and immunoblotting

Cells were harvested by scraping in chilled PBS and lysed in RIPA buffer with protease and protein phosphatase inhibitors. Protein concentration of cleared lysate was determined by BCA protein assay (Pierce). Lysates were separated in 4–12% NuPAGE gradient gels (Thermo Fisher), transferred to nitro-cellulose membrane and blocked with 5% milk in TBST using standard technique. Blocked membranes were immunoblotted with antibodies against the following targets separately: Phospho-PKA substrate (CST#9624), c-MYC (CST#18583), n-MYC (CST#84406), AURKA pThr288 (CST#3079), eIF4B pSer422 (CST#3591), eIF4B (CST#13088), GSK3B pSer9 (CST#9336), GSK3B (CST#12456), MAPK1/3 pThr202/Thr204 (CST#4370), MAPK1/3 (CST#4695), PKAC-α (CST#4782), PKAR1a (CST#5675), FLAG (Sigma#F1804), Actin (CST#3700), Vinculin (CST#13901), or COXIV (CST#5247). Afterward, blotted membranes were washed in TBST, incubated with appropriate HRP(Horseradish peroxidase)-labeled secondary antibodies (CST#7074, 7076), probed with ECL reagents (Thermo Fisher), and developed by x-ray. Blots were washed and stripped with Restore Plus stripping buffer (Thermo Fisher) if multiple probes were required. At least two distinct biological replicates were performed for any western blot analysis.

## Li-Cor western blot analysis

Cells were seeded, treated, harvested, separated by SDS-PAGE, and immunoblotted as described above, except Li-Cor specific secondary antibodies (CAT#926–32210, 926–32211) were used. The image was taken and quantified with the Li-Cor odyssey imaging system, and the half-life values were calculated using Prism.

## RNA sequencing

Total RNA was isolated using the Total Purification kit (Norgen Biotek, Thorold, ON, Canada). High capacity RNA to cDNA kit (Life Technologies, Grand Island, NY, USA) was used for reverse transcription of RNA. Libraries were generated by the Cornell Transcriptional Regulation and Expression (TREx) facility using the NEBNext Ultra II Directional Library Prep Kit (New England Biolabs, Ipswich, MA, USA) and subjected to paired-end sequencing on the NextSeq500 platform (Illumina) at the Genomics Facility in the Cornell University Biotechnology Resource Center. At least 80 M reads per sample were acquired. Reads were aligned to the human genome (build hg38) using STAR (*Dobin et al., 2013*) (v2.5.3a) for identification and quality control. Salmon (*Patro et al., 2017*) (v0.06.0) was used for transcript quantification with annotations from GENCODE release version 25. Normalization and differential gene expression analysis were carried out using DESeq2 (*Love et al., 2014*). Each of the samples had at least 25 million uniquely mapped reads and greater than 90% unique-mapping rate.

## Gene set enrichment analysis

GSEA for TCGA data was performed using the TCGA adrenocortical carcinoma (TCGA-ACC) and serous ovarian carcinoma (TCGA-OV) datasets, for which we obtained tumor somatic mutation and RNASEQ gene level read counts (normalized using the FPKM-UQ method) from the Genomic Data Commons Data Portal. There were 79 TCGA-ACC cases and 378 TCGA-OV cases for which RNASEQ data were available. For these cases, we ran GSEA (*Subramanian et al., 2005*) using default parameters and compared cases harboring PKAc amplifications/activating mutations to those without these alterations. For RNASeq data and proteomics data, we used the Enrichr web analysis tool (*Kuleshov et al., 2016*) to assess Hallmark Gene Sets (analysis performed 7/2022).

## Quantitative RT-PCR

Colo741 and FLX1 cells were seeded and treated in the same manner as described for immunoblotting in preparation for siRNA treatments. RNA was extracted with Trizol reagent (Thermo Fisher) according to the manufacturer's instructions and quantified with a NanoDrop instrument. Normalized RNA was reverse transcribed with SuperScript II Reverse Transcriptase (Invitrogen). cDNAs were added to PerfeCTa SYBR Green FastMix Reaction Mixes (QuantaBio) and respective primers and analyzed using the BioRad CFX Connect Real-Time PCR Detection System. Primers were designed with Primer3 and obtained from Elim Biopharmaceuticals (*Supplementary file 5*). Quality control was performed for each primer using amplification and melting curves. All experiments were done in at

least biological duplicate with three technical replicates per condition. If only one technical replicate did not show an appropriate amplification or melting curve, it was excluded from analyses.

## Cell viability assays

Cells were seeded into 96 well white opaque plates (Greiner) at 2000 cells per well and incubated at 37°C and 5% $CO_2$ overnight. Cells were treated with selected drugs at different final concentrations and incubated for another 72 hr except for the initial studies of Aurora Kinase inhibitors in FLX1, where incubation was 120 hr. After incubation, plates and CellTiter-Glo (CTG, Promega) reagent were allowed to equilibrate at room temperature on the bench for 30 min. The CTG assay was performed following the manufacturer's instructions and measured with a SpectraMax i3 Multi-Mode Platform (Molecular Devices). All experiments were done in at least biological duplicate with three technical replicates per condition. When multiple individual siRNA were used, the results are shown averaged in a standard dose response curve. In addition, AUC is calculated for each siRNA using GraphPad Prizm and shown as a separate point.

## Cell proliferation assays

For experiments with engineered FLX1 cells, 5333 cells of each line were seeded into black clear bottom 96 well plates (Corning) in 100 µl of media with or without dox (1 µg/ml). After seeding, plates were immediately incubated at 37°C and 5% $CO_2$ inside the Incucyte Zoom system (Essen BioScience) for live cell image and confluence analysis. For experiments with parental cells and siRNA, Colo741 and FLX1 cells were plated and treated with siRNA as described above. Cells were trypsinized after 24 hr of siRNA treatment and transferred to a black clear bottom 96 well plate at 500 cells per well. All experiments were done in at least biological duplicate with a minimum of three technical replicates per condition. The plates were allowed to incubate at 37°C and 5% $CO_2$ for 24 or 36 hr and moved to the Incucyte for further incubation. Once the plates were mounted inside the Incucyte system, pictures of each well were taken every 2 hr for confluence analysis.

## Kinase inhibitor library screening

FLX1 cells were seeded at 600 cells per well in 40 µl in 384 well plates. 5 µl of 10 µg/µl dox were added 24 hr after plating, and kinases inhibitors were added 48 hr after plating for a final concentration of 2 µM or 5 µM in total volume of 50 µl as listed. 5 d post inhibitor addition, cell viability of the cells were measured using Cell-Titer Glo as described above. The precise inhibitors screened are listed in *Supplementary file 5* and *Supplementary file 6*; they were purchased in library format from Selleck-Chem in 2017. For the data where FLX1 was exposed to the entire library without any genetic modification, statistical analysis was performed by developing a z-score within each screened plate and then averaging the z-scores across three biological replicates, allowing internal normalization. No samples or data points were excluded. For the data where FLX1 with dox-inducible *3xFLAG-PRKAR1A$^{G325D}$* was screened with the drug library ± dox treatment, a log(2)fold change vs. median was derived for each plate. This avoids amplifications in small differences in cell viability that may occur with a z-score. The normalized values ± dox were subtracted to generate an average log(2)fold change, and Student's t-test performed to determine statistical significance; given the relatively small number of compounds screened, no false discovery correction was used.

## siRNA kinase library screening

384 well plates containing the human protein kinase siGENOME siRNA library (Dharmacon Cat#G-003505) were thawed at room temperature and centrifuged at 1000 rpm for 5 min prior to foil removal. 50 µl of nuclease-free $dH_2O$ was added to each well to reconstitute the siRNA at a final concentration 5 µM. Using a Labcyte Echo 525 liquid handling machine, 200 nl of reconstituted siRNA from each well from the master plates was transferred to the same position of the corresponding black transparent bottom 384 well daughter plates (Thermo Fisher). Unused aliquoted plates were sealed with foil, covered with plastic lid, and stored at –80°C. For subsequent experiments, daughter plates with deposited siRNA were thawed at room temperature and centrifuged. 5 µl of nuclease-free $dH_2O$ was added to each well and agitated at room temperature for 30 min. 10 µl of a mixture of RNAiMAX and Opti-MEM was then added to each well and incubated at RT for 20 min. Finally, 500 FLX1 cells in 30 µl media were added to each well. The plates were transferred to the Incucyte for cell proliferation

monitoring. Statistical analysis was performed by developing a z-score within each screened plate and then averaging the z-scores across four biological replicates, allowing internal normalization. No samples or data points were excluded.

## TCGA analysis

TCGA PanCancer Project data between 3/13/18 and 4/23/18 were accessed through cBioPortal (at https://www.cbioportal.org) and queried by gene (e.g. *PRKACA* and *PRKAR1A*). Data were sorted through *Cancer Types Summary* function and exported to Microsoft Excel and Prism for reorganization and analysis.

## Phosphoproteomics

Engineered cell lines with dox-controlled 3xFLAG-*PRKACA* or *PRKAR1A*[G325D] constructs were treated with PBS or dox for 48 hr. Cells were then harvested in PBS, lysed in lysis buffer (8 µM urea, 50 mM Tris pH 8, 75 mM NaCl, and 1× protease and phosphatase inhibitors) and sonicated at 20% for 15 s. BCA protein assay was performed to quantify protein lysates. Samples were reduced with 5 mM dithiothreitol (DTT), cooled to room temperature, alkylated with 15 mM iodoacetamide, quenched with 15 mM DTT, diluted with 50 mM Tris pH 8 to <2 M urea, and subjected to trypsin digestion at 37°C overnight. Samples were acidified with 10% trifluoroacetic acid (TFA).

50 mg Seppak cartridges were set up on vacuum, and columns were washed with series of MS-grade acetonitrile (ACN), 70% ACN/0.25% acetic acid (AA), and 0.1% TFA buffers. After letting samples drip through columns, columns were washed with 0.1% TFA and 0.5% AA. Samples were eluted and lyophilized in a speed vacuum concentrator, and phosphopeptide enrichment was performed with immobilized metal affinity chromatography following established protocols (*Budzik et al., 2020*). Phosphopeptides were eluted in 50% ACN/0.1% formic acid (FA) and dried on a speed vacuum concentrator. Enriched samples were analyzed on a Q Exactive Orbitrap Plus mass spectrometry system (Thermo Fisher Scientific) with an Easy nLC 1200 ultra-high pressure liquid chromatography system (Thermo Fisher Scientific) interfaced via a Nanospray Flex nanoelectrospray source. Samples were injected on a C18 reverse phase column (25 cm × 75 µM packed with ReprosilPur C18 AQ 1.9 µM particles). Mobile phase A consisted of 0.1% FA and mobile phase B consisted of 80% ACN/0.1% FA. Peptides were separated by an organic gradient from 2 to 18% mobile phase B over 94 min followed by an increase to 34% B over 40 min, then held at 90% B for 6 min at a flow rate of 300 nl/min. MS1 data was acquired with a 3e6 AGC target, maximum injection time of 100 ms, and 70 K resolution. MS2 data was acquired for the 15 most abundant precursors using automatic dynamic exclusion, a normalized collision energy of 27, 1e5 AGC, a maximum injection time of 120 ms, and a 17.5 K resolution. All mass spectrometry was performed at the Thermo Fisher Scientific Proteomics Facility for Disease Target Discovery at UCSF and the J. David Gladstone Institutes.

Mass spectrometry data were assigned to human sequences, and peptide identification and label-free quantification were performed with MaxQuant (version 1.5.5.1) (*Tyanova et al., 2016*). Data were searched against the UniProt human protein database (downloaded 2017). Trypsin/P was selected allowing up to two missed cleavages. Standard quality control with variable modification was allowed for methionine oxidation, N-terminal protein acetylation, and phosphorylation of serine, threonine, and tyrosine, in addition to a fixed modification for carbamidomethyl cysteine. The other MaxQuant settings were left as default. Statistical analysis was performed using R (version 3.6.3), RStudio, and the MSstats Bioconductor package (*Choi et al., 2014*). These are broadly accepted statistical methods for mass spectrometry. Contaminants and decoy hits were removed, and samples were normalized across fractions by equalizing the median log2-transformed MS1 intensity distributions. Log2(fold change) for protein phosphorylation sites were calculated, along with p values. Phosphoproteomic data was uploaded to the PhosFate profiler tool (*Ochoa et al., 2016*; http://phosfate.com/) to infer kinase activity. Mass spectrometry RAW mass spectrum files are deposited into ProteomeXchange via PRIDE with the dataset identifier PXD025508.

## Multiplex inhibitor beads

MIBs were performed as described previously (*Donnella et al., 2018*; *Sos et al., 2014*). Kinase inhibitor compounds were purchased or synthesized and coupled to sepharose beads using 1-ethyl-3-(3-dimethylaminopropyl) carbodiimide chemistry. Engineered cell lines with dox-controlled *3xFLAG-PRKACA*

or *PRKAR1A*[G325D] constructs were treated with PBS or dox for 48 hr then collected in PBS. Samples were lysed in 150 mM NaCl buffer with protease and phosphatase inhibitors. Lysates were diluted with 5 M NaCl and high-salt binding buffer (50 mM Hepes pH 7.5, 1 M NaCl, 0.5% Triton X-100, 1 mM EDTA, and 1 mM EGTA). Pre-washed columns containing ECH sepharose 4B and EAH sepharose 4B beads were layered with kinase inhibitor-coupled beads as follows: 200 µl JG-4, 100 µl VI-16832, 75 µl staurosporin, 100 µl PP-hydroxyl, 100 µl purvalanol B, 50 µl AKTi-46, 100 µl dasatinib, 50 µl sorafenib, 50 µl crizotinib, 50 µl lapatinib, 50 µl SB202190, and 50 µl bisindolylmaleimide X. Columns were washed with high-salt buffer without disturbing bead layers, and affinity purification was performed with gravity chromatography. Bound kinases were washed with high-salt buffer, low-salt buffer (50 mM Hepes pH 7.5, 150 mM NaCl, 0.5% Triton X-100, 1 mM EDTA, and 1 mM EGTA), and 0.1% (w/v) SDS in high-salt buffer. Samples were eluted twice by capping the column, applying 300 µl of elution buffer (0.5% SDS/1% BME/0.1 M Tris-HCL pH 6.8) to the column, vortexing, heating to 98°C, removing caps, and allowing elution to flow through by gravity. Samples were frozen at –80°C overnight, reduced with 500 mM DTT, cooled to room temperature, and treated with 500 mM iodoacetamide. Methanol/ chloroform precipitation, trypsin digestion at 37°C overnight, and desalting were performed on all samples. Enriched samples were analyzed on a Q Exactive Orbitrap Plus mass spectrometry system (Thermo Fisher Scientific) with an Easy nLC 1200 ultra-high pressure liquid chromatography system (Thermo Fisher Scientific) interfaced via a Nanospray Flex nanoelectrospray source as described above for global phosphoproteomics. All mass spectrometry was performed at the Thermo Fisher Scientific Proteomics Facility for Disease Target Discovery at UCSF and the J. David Gladstone Institutes.

Peptides were identified with MaxQuant (version 1.5.5.1). Label-free quantification was performed with Skyline (*Schilling et al., 2012*), with Trypsin (KR|P) selected. Standard quality control was used, allowing up to two missed cleavages. Full scan MS1 filtering was performed with 70,000 resolving power at 400 m/z using the Orbitrap. Statistical analysis was performed with R, RStudio, and MSstats (*Choi et al., 2014*) to calculate log2(fold change) and p values of detected kinases. These are broadly accepted statistical methods for mass spectrometry. As above, mass spectrometry RAW mass spectrum files are deposited into ProteomeXchange via PRIDE with the dataset identifier PXD025508.

## Proteomics data integration and network propagation

Initial integration of the proteomics data was performed by identifying all kinases that were present in Phosfate or MIBs data from at least two cell lines. The abundance (MIBs) or imputed activity (Phosfate) was averaged between all cells with inducible *PRKACA* or *PRKAR1A*[G325D] and shown ± SD. For network propagation, the log(2)fold change values of MIBs data and effect size of Phosfate data for each engineered cell line treated with or without dox were separately normalized out of one. The union of these two datasets was generated, and any duplicate genes were averaged. Z-scores were then calculated, and the absolute values of the z-scores for each cell line were separately propagated using a random walk with restart (alpha = 0.2) across the ReactomeFI network using a MATLAB script available on github (*Huang et al., 2018*). Propagated heat scores for each gene were multiplied across cell lines containing the same construct (either dox-inducible *3xFLAG-PRKACA* or *PRKAR1A*[G325D]), and significance was calculated based on the probability that propagated heat scores match a permuted value by chance. Significant genes (p value<0.05) brought out by the network were then extracted and imported into Cytoscape (*Shannon et al., 2003*). To integrate engineered cell lines with Tet-on 3xFLAG-*PRKACA* or *PRKAR1A*[G325D], overlapping direct kinase neighbors of *PRKACA*, and their interconnections were extracted. The signs of the averaged z-scores of the Tet-on 3xFLAG-*PRKAR1A*[AG325D] lines were flipped and averaged with the averaged z-scores of the Tet-on 3xFLAG-*PRKACA* lines, resulting in a final subnetwork for PKAc. Nodes representing the genes were filled to represent the original z-scores which were averaged across cell lines. Networks were searched on Cytoscape for PKAc and its direct neighbors and any interconnections.

## Human FLC samples

Human FLCs and paired normal livers were obtained from the University of Washington Medical Center and Seattle Children's Hospital after institutional review board approval (Seattle Children's Hospital IRB #15277). For prospective fresh tissue collections, informed consent was obtained from the subject and/or parent prior to resection.

Fresh/frozen human FLC and paired non-tumor livers were homogenized in RIPA buffer with protease inhibitors using a hand-held Pro200 homegenizer (ProScientific). Protein concentration of cleared lysate was determined by BCA protein assay (Pierce). Lysate were separated by 10% TGX gels (Biorad), transferred to nitrocellulose membrane, and blocked with 5% milk in TBST using standard technique. Blocked membranes were immunoblotted with antibodies against following targets separately: PKAC-α (CST#4782), c-MYC (CST#18583), n-MYC (CST#84406), or Actin (Sigma#A5441). Afterward, blotted membranes were washed in TBST, incubated with appropriate HRP-labeled secondary antibodies (GE Healthcare Life Sciences), washed as before, and developed using ECL (Thermo Fisher) on an iBright FL1000.

## Materials availability

We will share all renewable reagents including plasmids, cell lines as well as assay methods, and protocols with the scientific community at large upon direct request to our laboratory. These include the FLX1 cell line (available under MTA) and all plasmids reported here.

## Data and code availability

All data generated or analyzed during this study are included in the manuscript and supporting files. Mass spectrometry RAW mass spectrum files have been deposited to the ProteomeXchange Consortium via the PRIDE partner repository with the dataset identifier PXD025508. Code used for network propagation is available on github as cited in the manuscript where it was initially described (*Huang et al., 2018*).

## Acknowledgements

This work was supported by the Fibrolamellar Cancer Foundation. Dr. Gordan is the recipient of a Burroughs Wellcome Career Award for Medical Scientists. Dr. Scott is supported by the National Institutes of Health (NIH) Grant DK119192, Dr. Yeung by the DOD CDMRP Grant# 12715138, Dr. Bouhaddou by NIH Grant F32CA239333, Dr. Krogan by NIH Grant U54 CA209891 and Dr. Turnham is supported by the National Center for Advancing Translational Sciences of the NIH Grant TR001871 and a Hope Funds for Cancer Research fellow, supported by the Hope Funds for Cancer Research (HCFR-21-05-05). We are grateful to all the patients and caregivers in the Fibrolamellar Liver Cancer community.

## Additional information

### Funding

| Funder | Grant reference number | Author |
| --- | --- | --- |
| Fibrolamellar Cancer Foundation | | John D Scott<br>John D Gordan<br>Nabeel Bardeesy |
| Burroughs Wellcome Fund Career Award | | John D Gordan |
| National Institutes of Health | DK119192 | John D Scott |
| DOD Peer Reviewed Cancer Research Program | 12715138 | Raymond S Yeung |
| National Institutes of Health | F32CA239333 | Mehdi Bouhaddou |
| National Institutes of Health | U54 CA209891 | Nevan J Krogan |
| Hope Funds for Cancer Research | HCFR-21-05-05 | Rigney E Turnham |

| Funder | Grant reference number | Author |
|---|---|---|
| National Center for Advancing Translational Sciences | TR001871 | Rigney E Turnham |

The funders had no role in study design, data collection and interpretation, or the decision to submit the work for publication.

## Author contributions

Gary KL Chan, Conceptualization, Data curation, Formal analysis, Investigation, Writing – original draft, Writing – review and editing; Samantha Maisel, Conceptualization, Data curation, Formal analysis, Investigation, Visualization, Writing – original draft; Yeonjoo C Hwang, Data curation, Software, Formal analysis, Investigation, Visualization, Methodology, Writing – original draft, Writing – review and editing; Bryan C Pascual, Rebecca RB Wolber, Huat C Lim, Donald Long, Investigation, Writing – review and editing; Phuong Vu, Investigation, Methodology; Krushna C Patra, Heidi L Kenerson, Praveen Sethupathy, Danielle L Swaney, Investigation, Methodology, Writing – review and editing; Mehdi Bouhaddou, Software, Methodology, Writing – review and editing; Raymond S Yeung, Conceptualization, Resources, Writing – review and editing; Nevan J Krogan, Resources, Methodology, Writing – review and editing; Rigney E Turnham, Conceptualization, Investigation, Methodology, Writing – review and editing; Kimberly J Riehle, John D Scott, Conceptualization, Resources, Methodology, Writing – original draft, Writing – review and editing; Nabeel Bardeesy, Conceptualization, Resources, Funding acquisition, Writing – original draft, Writing – review and editing; John D Gordan, Conceptualization, Data curation, Formal analysis, Funding acquisition, Investigation, Visualization, Methodology, Writing – original draft, Project administration, Writing – review and editing

## Author ORCIDs

Gary KL Chan http://orcid.org/0000-0002-8450-7831
Danielle L Swaney http://orcid.org/0000-0001-6119-6084
John D Scott http://orcid.org/0000-0002-0367-8146
John D Gordan http://orcid.org/0000-0001-8997-5725

## Ethics

Human subjects: Human FLCs and paired normal livers were obtained from the University of Washington Medical Center and Seattle Children's Hospital after institutional review board approval (SCH IRB #15277). For prospective fresh tissue collections, informed consent was obtained from the subject and/or parent prior to resection.

## Decision letter and Author response

Decision letter https://doi.org/10.7554/eLife.69521.sa1
Author response https://doi.org/10.7554/eLife.69521.sa2

# Additional files

## Supplementary files

- Supplementary file 1. Summary of TCGA analysis.
- Supplementary file 2. Individual phosphoproteomic, phosfate, and multiplex inhibitor beads (MIBs) datasets.
- Supplementary file 3. Network propagation results.
- Supplementary file 4. RNASEQ primary data.
- Supplementary file 5. Drug screen in FLX1.
- Supplementary file 6. Drug screen in FLX1 with dox-inducible PRKAR1A$^{G325D}$.
- Supplementary file 7. siKINOME final results.
- MDAR checklist

## Data availability

All data generated or analyzed during this study are included in the manuscript and supporting files. Mass spectrometry RAW mass spectrum files have been deposited to the ProteomeXchange

Consortium via the PRIDE partner repository with the dataset identifier PXD025508. The TCGA Adrenocortical Carcinoma and TCGA Ovarian Serous Cystadenocarcinoma datasets (https://www.ncbi.nlm.nih.gov/projects/gap/cgi-bin/study.cgi?study_id=phs000178.v11.p8) were used.

The following dataset was generated:

| Author(s) | Year | Dataset title | Dataset URL | Database and Identifier |
|---|---|---|---|---|
| Gordan | 2023 | Oncogenic PKA signaling stabilizes MYC oncoproteins via an aurora kinase A-dependent mechanism | https://www.ebi.ac.uk/pride/archive/projects/PXD025508 | PRIDE, PXD025508 |

The following previously published dataset was used:

| Author(s) | Year | Dataset title | Dataset URL | Database and Identifier |
|---|---|---|---|---|
| Weinstein JN, Collisson EA, Mills GB, Shaw KRM, Ozenberger BA, Ellrott K, Sander C, Stuart JM, Chang K, Creighton CJ, Davis C, Donehower L, Drummond J, Wheeler D, Ally A, Balasundaram M, Birol I, Butterfield YSN, Chu A, Kling T | 2013 | The Cancer Genome Atlas (TCGA) | https://www.ncbi.nlm.nih.gov/projects/gap/cgi-bin/study.cgi?study_id=phs000178.v11.p8 | dbGap, phs000178 |

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
