## [Editor Report]

The authors employed global kinome profiling to identify key effectors of protein kinase A (PKA) oncogenic signalling in fibrolamellar carcinoma and melanoma cell line models. Based on subsequent cell line-based validation using standard molecular and cellular biology assays, authors propose a model whereby the oncogenic effects of PKA are at least in part mediated by c-MYC. In addition to stabilizing c-MYC protein, the authors provide some evidence that PKA may stimulate c-MYC protein synthesis in an eukaryotic translation initiation factor 4F (eIF4F)-dependent manner. Notwithstanding that the underlying mechanisms remain obscure, it was thought that this study is of broad interest inasmuch as it provides hitherto unacknowledged insights into the molecular underpinnings of oncogenic PKA signalling and accordingly, it was thought that this manuscript may be of interest to researchers in the fields of cancer research, therapeutics, signal transduction and molecular and cell biology.

---

## [Decision Letter]

**Decision letter after peer review:**

Thank you for submitting your article "Oncogenic PKA signaling stabilizes MYC oncoproteins via an aurora kinase A-dependent mechanism" for consideration by *eLife*. Your article has been reviewed by 3 peer reviewers, including Ivan Topisirovic as the Reviewing Editor and Reviewer #1, and the evaluation has been overseen by Erica Golemis as the Senior Editor.

Essential revisions:

1. Relative lack of data regarding the signaling mechanisms to corroborate the proposed model was found to be a major drawback of this study. In particular, it was thought that additional mechanistic evidence linking oncogenic PKA signaling to PIM activity is warranted. In addition, it should be established whether the activation of AURKA by PKA occurs directly or indirectly (e.g. via the effects of PKA on cell proliferation). Addressing these issues is required to support key conclusions of the article.

2. Evidence for the role of MYC family members as key effectors of PKA signaling in neoplasia was deemed to be insufficient. Additional experiments are required to firmly establish that the levels of MYC family members are indeed regulated via the PKA-AURKA/PIM axis and not secondary to the effects of modulation of PKA signaling on the proliferative state of the cell. In addition, alternative mechanisms that may underpin the effects of PKA on c-MYC and n-MYC protein levels (e.g. alterations in translation of corresponding mRNAs) should be considered. Finally, the mechanism(s) whereby AURKA and PIM regulate MYC family member levels remain largely elusive. Overall, it was thought that additional mechanistic evidence related to MYC regulation in the context of constitutive PKA activation is merited.

3. Some methodological problems were observed. Specifically, several key experiments rely on a single siRNA and/or pharmacological inhibitor. Orthogonal approaches, employing additional siRNAs, appropriate rescues, and/or MYC mutants are advised.

4. The cancer relevance of results obtained in the studies that relied on the ectopic expression of proteins is questionable. These concerns were based on apparent discrepancies between AURKA and PIM2 levels in FLC tumor lesions and the lack of their correlation with c-MYC and n-MYC expression. In addition, observed variability between AURKA and PIM2 expression between tumor and adjacent liver suggest potential additional PKA-dependent mechanisms of MYC regulation. This, in conjunction with the absence of in vivo studies, suggests that the authors should consider toning down claims regarding clinical translatability of their findings.

*Reviewer #1 (Recommendations for the authors):*

– As authors pointed out, in figure 6B there was a high variability between AURKA and PIM2 expression between tumor and adjacent liver, thus suggesting that additional mechanisms of PKA-dependent MYC stabilization may be in play in vivo. Furthermore, the most of experiments were done in cell lines and thus do not represent what may be transpiring under physiological conditions. To this end, it appears to be warranted that the authors test their model in vivo (e.g. by studying the effects of AURKA inhibitors in PKA-driven cancer xenograft models).

– Mechanistically, it remains largely unclear how is PKA signaling linked to PIM activity. Although it is reasonable that precise mechanistic dissection may be outside of the scope of the present manuscript, some additional mechanistic experiments are warranted to support the correlative data provided by the authors. Moreover, it is not clear why there is a discrepancy in the effects of PIM1 vs. PIM2 in Colo741 vs. FLX cells, and what was the motivation to pursue PKA effects on PIM2 in FLX1 cells, wherein PIM1, but not PIM2 depletion resulted in reduction of c-MYC levels.

– Validation of the phosphoproteome and kinome activity data relies heavily on pharmacological approaches. It was thought that orthogonal genetic approaches (and possibly some longer lasting than using siRNA) are merited to further corroborate the authors model.

– There is appreciable difference between the effects of AURKA inhibitors on MYC between Colo741 and FLX1 cells, whereby in the latter case the effects especially with the MLN compound are not very convincing. The authors should comment on this.

– Experiments employing non-degradable MYC mutants to rescue the effects of DN PRKACA overexpression on e.g. proliferation and clonogenic growth seem to be appropriate to firmly establish the extent to which the effects of PRKACA are mediated via MYC.

– In figure 1D siRNA approach was used for a clonogenic assay. The latter assay takes relatively long time compared to relatively transient effects of siRNA. What was the expression of PKA at the end point of the clonogenic assay? Also, the control for the efficiency of PRKACA siRNA should be included. Similar controls also appear to be missing in figure 1E.

– The authors should consider improving the description of figure 1G, as in the text it is not indicated that expression of DN PRKACA was compared to the overexpression of WT PRKACA in the left panel.

– As far as I know, the origin of Colo741 cell lines is still disputed. The authors should note this in the text.

– Methodology was not described in sufficient detail. More detailed information how phosphoproteome and proteome data were analyzed and integrated would be appreciated. In particular in some parts of the paper it is not clear what statistical analyses were employed. This should be clearly indicated in figure legends and methods section.

– This is a comment that is outside the scope of the article (comes purely from the enthusiasm of the reviewer for this study), but considering initial findings of the authors implicating LKB1 it may be pertinent to test the effects of biguanides and mTOR inhibitors in experiments presented in figure 3A. Perhaps simultaneous use of these compounds with AURKA inhibitors should also be considered.

*Reviewer #2 (Recommendations for the authors):*

1. Cause or Consequence Issue: One of the main issues is that the regulation of MYC and/or N-MYC may be caused by PKA signaling, as proposed, but may also be an indirect consequence of the growth state of the cell. MYC is highly regulated with a ~30 min half-life that is highly responsive to signaling pathways controlling cell proliferation. To distinguish this 'cause or consequence' issue the authors must measure the growth state of the cell in response to PKA activation/inhibition, by conducting cell cycle profiling for example, and show that the regulation of MYC occurs prior to the effect on cell growth. This will involve kinetic analysis of MYC expression at short intervals following PKA activation/inhibition.

2. PKA regulation of protein translation vs MYC stability per se: Regulation of MYC stability is often stated throughout the manuscript (e.g. line 207; Figure 4), yet MYC half-life experiments were not performed. Is the net result of PKA inactivation a 'change' in MYC half-life or is MYC translation blocked and the decay of existing MYC pools then dissipating with the usual ~30 min half-life? Perhaps the substrates of PKA signaling that contribute to growth occurs at the level of protein translation. What happens to other short half-life proteins in the cell in response to PKA regulation (e.g. MCL^-^1)? Is this PKA-mediated effect on protein stability MYC specific?

3. MYC phosphorylation and degradation: The authors draw MYC interaction with AURKA and PIM kinases (Figure 5G) suggesting these kinases directly phosphorylate MYC. Further evidence that these kinases regulate MYC directly would add weight to the model proposed. Phosphosite plus lists many sites phosphorylated on MYC and N-MYC, yet I don't think any of these have been ascribed to AURKA and/or PIM. One well-studied phosphosite is Serine 62, which is phosphorylated by mitogen stimulated kinases to promote MYC activity and cell growth. It is also conserved between MYC and N-MYC. Is this the site of phosphorylation by AURKA and/or PIM? Or is this the role of the RAS/MAPK pathway downstream of PKA? Serine 62 is a priming site for GSK-3 to phosphorylate Threonine 58, which then recruits SCF(Fbxw7) to degrade MYC so the Serine 62 site is a high-probability candidate residue for PKA regulation by one or both of these downstream pathways. Interestingly, GSK3 is in the same node as AURKA in the authors analyses (Figure 2E), although α is increased and β decreased. Further interrogation on the site of phosphorylation would add weight to the authors model. Determining the role of the GSK-3-SCF(Fbxw7) in PKA regulation of MYC is important and can be achieved by knocking out Fbxw7 and then determining whether MYC expression continues to be regulated by PKA.

4. Only one siRNA used throughout: it is the standard in the field to evaluate more than on siRNA per gene of interest to ensure effect is specific and not due to off-target effects. For example, see Figure (3D, F, 5B, 5D, etc.)

5. Lines 186-191: Are cells dependent on PKA also dependent on MYC?

What is the dependency score as reported in the Cancer Dependency Map for MYC in the Colo741 cells as they were 'highly dependent on PKA (line 104)' ? Provide relative dependency scores of PKA and MYC in the cell lines used in this study and across all solid cancers.

6. Kinome-wide siRNA library screen: The results of this screen need to be provided as a supplementary table so hit analysis can be reviewed by the reader. Did the expected positive controls come up as hits? For example, the FLX1 cells are grown in HGF, so a positive control hit would be HGFR (Met). Was AURKA a hit?

7. Data interpretation (Figure 5C): In Colo741 cells MLN8237 appears to decrease MYC expression more than the MLN8237 and CS6258 together. It is more like an additive effect, leading to intermediate levels of MYC. C- and N-myc levels are so low in the FLX1 cells it is difficult to really see any significant decrease. Densitometry would help score the response to drug.

8. Data weak (Figure 5D): Decrease of PIM2 is not evident in Colo741 as described in text for Figure 5D. Try stimulating PKA with FSK/IMBX (as in Figure 4) in the presence and absence of siRNA to PRKACA.

9. Use of AML12: This transgenic knockin of DNAJB1-PRKACA in murine hepatocytes may be the 'cleanest' cell system to interrogate the proposed PKA-AURKA/PIM-MYC axis as the only genetic alteration is the transgene (Figure 6A).

10. Figure 6B: Many mechanisms regulate MYC expression. It remains unclear whether the PKA-AURKA/PIM-MYC axis is functional in FLC as described as Figure 6B not convincing.

*Reviewer #3 (Recommendations for the authors):*

The study by Chan, Gordan et al., utilizes state of the art global kinome profiling to map the shared signaling networks driven by diverse genetic changes resulting in PKA activation in human cancer. Among the many kinases whose activity is modulated by active PKA, they authors centered their study on Aurora Kinase A (AURKA), and its ability to regulate c-MYC and n-MYC protein levels. They propose an AURKA-MYC regulatory network, and a possible positive feedback loop mediated by the kinase PIM2, which can be disrupted by AURKA inhibition. The study has many elements of novelty, which could be of translational importance. The strength of the study is the use of global kinome profiling and the identification of multiple candidate signaling nodes downstream from PKA. The weakness is the limited mechanistic information provided on possible direct regulatory processes intervening in kinase activation, and the need to enhance the rigor of the studies and to provide quantitative analysis of the data to increase the confidence regarding the proposed novel mechanisms at this stage.

What is the mechanism by which PKA activates AURKA? Is AURKA phosphorylated by PKA? AURKA expression levels and enzymatic activity changes occur with the cell cycle, so it is unclear what the direct link between PKA and AURKA is, or if this (and other) effects are due to the impact of PKA on general cell growth or other indirect processes.

Similarly, the proposed link between PKA-AURKA, MYC, and the positive feedback through PIM2 is not mechanistically defined.

Figure 1:

The frequency (not the incidence?) of PRKACA amplification described in the text (0.3-3.2%) does not match with the information in the corresponding figure (1B).

With all the emphasis on the validity of FLX1 cells as a model for PKA-driven FLC, it is not clear why these cell were not used for proteomics analysis (Panel C).

Unless mistaken, Colo741 skin cancer cells have a frameshift mutation in PRKAR1, but the expression levels of this protein did not change (1G, third and fourth lane).

There seems to be little expression of PKA C subunit in ML1 cells after dox induction.

Figure 2:

The kinases whose activity is regulated by PKA C expression may be direct or indirect, and hence caution may need to be taken regarding the direct regulatory role of PKA on these kinases. For example, the authors mention that many of these kinases are involved in G2M, and hence it is possible that PKA C regulates these kinases indirectly through promoting cell cycle progression, among other indirect mechanisms.

In Sup 1, ML1 cells that are not sensitive to PKA inhibition were used, without a clear explanation. That said, cell proliferation inhibition by MEK1/2 and ERK1/2 inhibitors were less sensitive in these cells, and mentioned as probably of no interest, whereas there was a very limited increase in the sensitivity to one AURKAi, CD532, and changes in the responses to another MLN8237, which itself was not very effective, but the latter deemed of importance. This reviewer is unclear about the rationale and biological interpretation of the data. The comment on "older generation" AURKAi (ENMD-2076) affecting PKA C induction appears to be odd (line 172).

Figure 3:

Seems that only CD532 is effective in reducing proliferation of PKA C dependent cells among all AURKAi tested, raising the possibility of an off target effect.

In that venue, changes in gene expression reported, most of which are involved in cell cycle progression, may be due to decrease cell proliferation and not acting directly downstream from PKA C, including c-MYC.

Data on C-MYC siRNA effects (and in fact all siRNA studies, including MYCN) require more than one siRNA and quantitative analysis of the impact on protein expression and changes.

This also applies to cell proliferation. For example, in Colo741 cells there appears to be less than 20% decrease in cell proliferation after MYC siRNA use, which is of questionable biological relevance.

Information of cell groups (grey/red/yellow) in FLX1 experiments is missing and cannot be reviewed.

The text on c-MYC and n-MYC knock down in Colo741 cells seems to be incorrect (line 187) as these cells do not express n-MYC. The impact of n-MYC siRNA on the protein expression of n-MYC (Figure 3D) is not clear, and needs quantification as for most other similar knockdown studies.

Figure 4:

Panel B. As for other similar studies, need to use of more than 1 siRNA for PRKACA to decrease the possibility of off-target effects.

Figure 5.

Panel B. Use more than one siRNA for PIM1 and PIM2. This is emphasized by the fact that PIM2 silencing abrogates the expression of both PIM proteins.

The use of MLN8237 and CX6258 does not seem to provide "cooperating" or even additive effects with respect to c-MYC and n-MYC expression.

The positive feedback loop mediated by PIM2 downstream AURKA and MYC appears to be speculative, and not mechanistically explained, at the protein interaction, activity changes, and/or expression level.

Figure 6:

AURKA and PIM2 levels are not regulated in FLC tumor lesions harboring endogenous PKA C activating protein as proposed by the studies using ectopically expressed proteins, nor do they appear to correlate with c-MYC and n-MYC expression levels. This raises concerns about the overall cancer relevant significance of the proposed regulatory model in this study.

---

## [Author Response]

Essential revisions:1. Relative lack of data regarding the signaling mechanisms to corroborate the proposed model was found to be a major drawback of this study. In particular, it was thought that additional mechanistic evidence linking oncogenic PKA signaling to PIM activity is warranted. In addition, it should be established whether the activation of AURKA by PKA occurs directly or indirectly (e.g. via the effects of PKA on cell proliferation). Addressing these issues is required to support key conclusions of the article.

As described in our summary, we identified altered translation as the likely primary mechanism of PKA-induced increases in c-MYC during the revision process. As a result, we have shifted the focus of this manuscript away from the AURKA/PIM signaling axis towards effects on protein translation. However, we also include new data showing that PKA-induced increases in c-MYC protein level occur in cells that are synchronized into G2/M with nocodazole (Figure 5—figure supplement 1B), mitigating the concern that alterations in cell cycle distribution underly PKA-induced c-MYC changes.

We agree that the mechanistic connection between PKA and PIM is not well substantiated. Unfortunately, there are only limited reagents available to study PIM kinases directly, and their substrates are poorly defined and/or shared with many other kinases. While our revision includes our data showing PIM2 inhibitor effects on c-MYC protein levels, we have removed the data connecting PIM2 to PKA activity and do not make any claims that it is activated by PKA.

2. Evidence for the role of MYC family members as key effectors of PKA signaling in neoplasia was deemed to be insufficient. Additional experiments are required to firmly establish that the levels of MYC family members are indeed regulated via the PKA-AURKA/PIM axis and not secondary to the effects of modulation of PKA signaling on the proliferative state of the cell. In addition, alternative mechanisms that may underpin the effects of PKA on c-MYC and n-MYC protein levels (e.g. alterations in translation of corresponding mRNAs) should be considered. Finally, the mechanism(s) whereby AURKA and PIM regulate MYC family member levels remain largely elusive. Overall, it was thought that additional mechanistic evidence related to MYC regulation in the context of constitutive PKA activation is merited.

We appreciate this insight. In addressing these concerns, we find that while AURKA and PIM2 do regulate MYC levels in PKA-driven cancers, the key influence is in fact at the level of translation as proposed by the reviewers. Substantial new data have been added to support this observation, with additional investigation of cell cycle dependence on PKA effects on MYC (Figure 4—figure supplement 1B and Author response image 1) and PKA effects on translation and MYC, and the use of relevant mutant constructs (Figure 6-7; Figure 5—figure supplement 1; Figure 7—figure supplement 1). We further provide data from TCGA that MYC transcriptional targets are upregulated in the presence of PKA-activating mutations, unlikely to be solely due to cell cycle effects. Specific results are detailed in the response to reviewers.

**Author response image 1. sa2fig1:** PKA effects on cell cycle distribution in FLX1. (A) FLX1 parental cells were treated with DMSO or 50 μM FSK/IBMX for 4 hours, with BrdU added for the last 20 minutes. Cells were then stained for nuclear content with 7-AAD and active DNA synthesis with anti-BrdU antibody, then acquired by FACS. Average of 3 samples is shown. Statistically significant increase in S-phase% with FSK/IBMX was noted by one-tailed Student’s t-Test (p = 0.002). (B) FLX1 cells with dox-inducible *3xFLAG-PKAR1AG325D* were treated with ±dox for 48 hours and then analyzed as above. Statistically significant decrease in S-phase% with dox was noted by one-tailed Student’s t-Test (p = 0.03). We note the markedly low S-phase percentage shown here. FLX1 cells have a >72 hour doubling time, and the same labeling/staining conditions show significantly higher S-phase% in many other tumor cell lines in our hands.

3. Some methodological problems were observed. Specifically, several key experiments rely on a single siRNA and/or pharmacological inhibitor. Orthogonal approaches, employing additional siRNAs, appropriate rescues, and/or MYC mutants are advised.

As recommended, we have added individual siRNA analysis, additional compounds and the use of MYC mutants throughout the manuscript.

4. The cancer relevance of results obtained in the studies that relied on the ectopic expression of proteins is questionable. These concerns were based on apparent discrepancies between AURKA and PIM2 levels in FLC tumor lesions and the lack of their correlation with c-MYC and n-MYC expression. In addition, observed variability between AURKA and PIM2 expression between tumor and adjacent liver suggest potential additional PKA-dependent mechanisms of MYC regulation. This, in conjunction with the absence of in vivo studies, suggests that the authors should consider toning down claims regarding clinical translatability of their findings.

This point is well taken. The objective of this manuscript is to map the multiple signaling mechanisms downstream of oncogenic PKA, and we acknowledge that further clinical validation is required. We have altered the text of the discussion to further emphasize this point. In addition, the manuscript has been reorganized and data regarding AURKA and PIM2 in tumor specimens was removed. We have also strengthened the data in support of the relevance of PKA activation to MYC transcriptional activity in patient genomic datasets (Figure 3F).

Reviewer #1 (Recommendations for the authors):– As authors pointed out, in figure 6B there was a high variability between AURKA and PIM2 expression between tumor and adjacent liver, thus suggesting that additional mechanisms of PKA-dependent MYC stabilization may be in play in vivo. Furthermore, the most of experiments were done in cell lines and thus do not represent what may be transpiring under physiological conditions. To this end, it appears to be warranted that the authors test their model in vivo (e.g. by studying the effects of AURKA inhibitors in PKA-driven cancer xenograft models).

We fully acknowledge this point. We have restructured the manuscript so that the primary in vivo data relate to the expression and transcriptional activity of MYC proteins in PKA-driven cancers (Figure 3F-G). The ensuing in vitro analysis delineates several mechanisms by which PKA can regulate c-MYC and posits that translational regulation is the most important. With the scope of this additional work, and relative difficulties in obtaining clinically relevant model systems and compounds, we feel that additional therapeutic studies to are outside of the scope of this manuscript. We state these as key next steps and plan to undertake them in future work.

– Mechanistically, it remains largely unclear how is PKA signaling linked to PIM activity. Although it is reasonable that precise mechanistic dissection may be outside of the scope of the present manuscript, some additional mechanistic experiments are warranted to support the correlative data provided by the authors. Moreover, it is not clear why there is a discrepancy in the effects of PIM1 vs. PIM2 in Colo741 vs. FLX cells, and what was the motivation to pursue PKA effects on PIM2 in FLX1 cells, wherein PIM1, but not PIM2 depletion resulted in reduction of c-MYC levels.

We acknowledge the reviewer’s point. Unfortunately, the limited reagents and knowledge of PIM-specific functions has proven a challenge in our analysis of the potential connection between PKA, MYC and PIM. Accordingly, we have de-emphasized these data in the manuscript in favor of more expansive work focusing on the eIF4F complex. We have also removed several of these figures as we do not believe that they contribute significantly to the story and may distract from key points.

– Validation of the phosphoproteome and kinome activity data relies heavily on pharmacological approaches. It was thought that orthogonal genetic approaches (and possibly some longer lasting than using siRNA) are merited to further corroborate the authors model.

Additional direct validation of our proteomics results have been added, both with signaling analysis (Figure 2F) and through an integrated pharmacological/genetic analysis in our FLC cell model (Figure 4B). We note some variability in signaling between the cell lines studied in 2F, but this is anticipated given their distinct genetic backgrounds. The PKA inhibiting tool compound H89 was also used. We agree that longer lasting genetic strategies (e.g. CRISPRi) would strengthen the manuscript. This was vigorously attempted, but unsuccessful. Unfortunately, the FLX1 cell model was not tolerant of chronic knockdown of the targets under study. We note this as a specific limitation of the study in our discussion

– There is appreciable difference between the effects of AURKA inhibitors on MYC between Colo741 and FLX1 cells, whereby in the latter case the effects especially with the MLN compound are not very convincing. The authors should comment on this.

Additional comments have been added to the results and discussion reflecting this point.

– Experiments employing non-degradable MYC mutants to rescue the effects of DN PRKACA overexpression on e.g. proliferation and clonogenic growth seem to be appropriate to firmly establish the extent to which the effects of PRKACA are mediated via MYC.

The use of genetic models of MYC stabilization, either with constructs lacking a 5’UTR or with the T58A mutation have been added. Constructs are used in Figure 7 and Figure 7—figure supplement 1A-B to confirm that the effects of PKAc knockdown and eIF4A inhibition are abrogated in when MYC lacks a 5’ untranslated region (UTR). Given our new findings, non-degradable MYC alleles such as T58A are no longer as relevant to the focus of the story but is included in an additional figure for reviewers’ interest (Author response image 2). An experiment combining PKA inhibition and an insensitive variant of c-MYC (lacking a 5’UTR) is included (Figure 7—figure supplement 1A) using pooled siRNA in place of DN-PKA.

**Author response image 2. sa2fig2:** FLX1 and Colo741 with dox-induced *3xFLAG-c-MYC* lacking a 5’UTR and including the stabilizing T58A mutation. Dox treatment was 1 μG/mL for 40 hours prior to harvest with zotatifin or DMSO for 24 hours. Of note, the *MYCT58A* allele was toxic in both cells but particularly in FLX1, resulting in reduced expression of vinculin and other housekeeping proteins.

– In figure 1D siRNA approach was used for a clonogenic assay. The latter assay takes relatively long time compared to relatively transient effects of siRNA. What was the expression of PKA at the end point of the clonogenic assay? Also, the control for the efficiency of PRKACA siRNA should be included. Similar controls also appear to be missing in figure 1E.

This point is well taken. We have removed the clonogenic assay and added multiple assays of PKA-dependent proliferation in FLX1 with appropriate controls (Figure 3—figure supplement 1A-B).

– The authors should consider improving the description of figure 1G, as in the text it is not indicated that expression of DN PRKACA was compared to the overexpression of WT PRKACA in the left panel.

The description of this figure has been clarified.

– As far as I know, the origin of Colo741 cell lines is still disputed. The authors should note this in the text.

This point is now included in the text (line 108-109).

– Methodology was not described in sufficient detail. More detailed information how phosphoproteome and proteome data were analyzed and integrated would be appreciated. In particular in some parts of the paper it is not clear what statistical analyses were employed. This should be clearly indicated in figure legends and methods section.

An additional section has been added to the methods section to explain our data integration in Figure 2C-D (lines 643-646), and comments added to the figure legends clarifying what statistical tests were used in each case.

– This is a comment that is outside the scope of the article (comes purely from the enthusiasm of the reviewer for this study), but considering initial findings of the authors implicating LKB1 it may be pertinent to test the effects of biguanides and mTOR inhibitors in experiments presented in figure 3A. Perhaps simultaneous use of these compounds with AURKA inhibitors should also be considered.

We appreciate and share the reviewer’s interest in the LKB1 result. This is of particular interest given that the PKA-inhibited SIK kinases are direct LKB1 substrates. Our new data in Figure 5B speak to this question, with reduced sensitivity to PI3K/mTOR inhibition when PKA is inhibited, potentially due to reduced mTOR pathway signaling as a result of LKB1 activation. We look forward to investigating this relationship in future work.

Reviewer #2 (Recommendations for the authors):1. Cause or Consequence Issue: One of the main issues is that the regulation of MYC and/or N-MYC may be caused by PKA signaling, as proposed, but may also be an indirect consequence of the growth state of the cell. MYC is highly regulated with a ~30 min half-life that is highly responsive to signaling pathways controlling cell proliferation. To distinguish this 'cause or consequence' issue the authors must measure the growth state of the cell in response to PKA activation/inhibition, by conducting cell cycle profiling for example, and show that the regulation of MYC occurs prior to the effect on cell growth. This will involve kinetic analysis of MYC expression at short intervals following PKA activation/inhibition.

We are grateful for this insightful recommendation. The kinetic analysis of c-MYC expression is included (Figure 5—figure supplement 2C), and demonstrates no change in c-MYC stability following PKA stimulation. c-MYC levels were largely abolished by PKA inhibition, so its stability could not be tested in that context. In addition to the proliferation data that have been added to this manuscript, we have tested the impact of FSK/IBMX and PRKAR1A^G325D^ induction on cell cycle progression in FLX1. These cells are markedly slow growing, with approximately 1% of cells labeling with BrdU in conditions that label ~20% of cells acquired for another experiment in parallel. While statistically significant changes in % of cells labeling with BrdU were noted, we do not feel that they are adequate to explain the increase in c-MYC protein levels (Author response image 1). This is supported by our testing of c-MYC mRNA (Figure 3B), which shows a doubling in mRNA in FLX1 but not Colo-741, despite both cell lines showing increased protein levels. Similarly, synchronization with nocodazole did not block PKA’s induction of c-MYC expression. We acknowledge that altered growth rates/cell cycle status may contribute to c-MYC increases, but these many lines of evidence support a model where PKA increases c-MYC levels in part via direct effects.

2. PKA regulation of protein translation vs MYC stability per se: Regulation of MYC stability is often stated throughout the manuscript (e.g. line 207; Figure 4), yet MYC half-life experiments were not performed. Is the net result of PKA inactivation a 'change' in MYC half-life or is MYC translation blocked and the decay of existing MYC pools then dissipating with the usual ~30 min half-life? Perhaps the substrates of PKA signaling that contribute to growth occurs at the level of protein translation. What happens to other short half-life proteins in the cell in response to PKA regulation (e.g. MCL^-^1)? Is this PKA-mediated effect on protein stability MYC specific?

As alluded to above, this valuable recommendation has been acted on. We completed the proposed experiment (Figure 5—figure supplement 2C), showing that MYC stability is in fact not decreased by PKA. This analysis motivated our study of protein translation and the inclusion of multiple new figures connecting PKA and protein translation (Figure 6, 7; Figure 7—figure supplement 1). The text has been modified as well accordingly. Given our finding that c-MYC stability is not altered, we did not assess the stability of other short half-life proteins. However, we do include for the reviewer’s interest evidence that protein levels of another translationally regulated protein (ERBB2, see Gerson-Gewirtz et al., 2021; manuscript reference 57) are also increased by PKAc stimulation and blocked by the eIF4A inhibitor zotatifin (Author response image 3).

**Author response image 3. sa2fig3:** FSK/IBMX and zotatifin effects on other translationally regulated transcripts. (A) Time course of FSK/IBMX in Colo741 and FLX1 showing increased expression of ERBB2; kinetics are somewhat different between the two cell lines. (B) Effect of 24 hour treatment with 100 nM zotatifin on ERBB2 protein levels performed on same samples as shown in Figure 7C.

3. MYC phosphorylation and degradation: The authors draw MYC interaction with AURKA and PIM kinases (Figure 5G) suggesting these kinases directly phosphorylate MYC. Further evidence that these kinases regulate MYC directly would add weight to the model proposed. Phosphosite plus lists many sites phosphorylated on MYC and N-MYC, yet I don't think any of these have been ascribed to AURKA and/or PIM. One well-studied phosphosite is Serine 62, which is phosphorylated by mitogen stimulated kinases to promote MYC activity and cell growth. It is also conserved between MYC and N-MYC. Is this the site of phosphorylation by AURKA and/or PIM? Or is this the role of the RAS/MAPK pathway downstream of PKA? Serine 62 is a priming site for GSK-3 to phosphorylate Threonine 58, which then recruits SCF(Fbxw7) to degrade MYC so the Serine 62 site is a high-probability candidate residue for PKA regulation by one or both of these downstream pathways. Interestingly, GSK3 is in the same node as AURKA in the authors analyses (Figure 2E), although α is increased and β decreased. Further interrogation on the site of phosphorylation would add weight to the authors model. Determining the role of the GSK-3-SCF(Fbxw7) in PKA regulation of MYC is important and can be achieved by knocking out Fbxw7 and then determining whether MYC expression continues to be regulated by PKA.

We appreciate this insightful point as well. As our pursuit of this reviewers’ recommendation 2 moved the manuscript away from a focus on MYC degradation, we did not pursue detailed mechanistic analysis to link activated kinases to specific c-MYC phosphosites. We do note that there were no significant alterations in phosphorylation of any MYC phosphosites in our proteomic data sets. In addition, further experiments studying PKA effects on AURKA and GSK3, as well as GSK3 effects on MYC expression (Figure 5—figure supplement 1C) and the impact of proteosome inhibition were included (Figure 5—figure supplement 2). Mutant forms of MYC lacking a 5’UTR are studied in Figure 7 and Figure 7—figure supplement 1. The T58A allele is not studied in the main text but is included for the reviewer’s interest (Author response image 2).

4. Only one siRNA used throughout: it is the standard in the field to evaluate more than on siRNA per gene of interest to ensure effect is specific and not due to off-target effects. For example, see Figure (3D, F, 5B, 5D, etc.)

We apologize for the lack of clarity in our prior submission – pooled siRNA were used in all cases. In response to the reviewers’ concerns, single siRNAs have been substituted for mRNA expression analysis and proliferation analysis as advised. Pooled siRNAs are still used for some protein experiments as the knockdown achieved with single siRNA was often limited in larger scale cellular assays. We have complemented the siRNA experiments with additional compounds to overcome this limitation.

5. Lines 186-191: Are cells dependent on PKA also dependent on MYC?What is the dependency score as reported in the Cancer Dependency Map for MYC in the Colo741 cells as they were 'highly dependent on PKA (line 104)' ? Provide relative dependency scores of PKA and MYC in the cell lines used in this study and across all solid cancers.

The DepMap reports the following:

**Author response table 1. sa2table1:** 

	CRISPR	
	PRKACA	MYC
Average	-0.24	-1.46
639V	-0.14	-0.67
Colo741	-0.35	-1.38
		
	siRNA	
	PRKACA	MYC
Average	-0.15	-0.48
Colo741	-1.38	-0.35

Unfortunately, ML-1 was not included in these data sets. We note that MYC is a common essential gene and many cell lines demonstrate dependency on MYC when tested by CRISPR. Colo741 is one of the most PKA dependent cells in both the CRISPR and siRNA datasets. We have amended the text to make this point more completely.

6. Kinome-wide siRNA library screen: The results of this screen need to be provided as a supplementary table so hit analysis can be reviewed by the reader. Did the expected positive controls come up as hits? For example, the FLX1 cells are grown in HGF, so a positive control hit would be HGFR (Met). Was AURKA a hit?

The results of this screen are included as a data supplement. AURKA in fact was not a hit, a point that we have highlighted in the text. *PRKACA* serves as a positive control and is labeled in the figure. We note that common essential genes WEE1 and PLK1 each have a z-score of -1, slightly greater that PRKACA but less than other genes highlighted in the figure. *MET* has a z-score of -0.62.

7. Data interpretation (Figure 5C): In Colo741 cells MLN8237 appears to decrease MYC expression more than the MLN8237 and CS6258 together. It is more like an additive effect, leading to intermediate levels of MYC. C- and N-myc levels are so low in the FLX1 cells it is difficult to really see any significant decrease. Densitometry would help score the response to drug.

We agree with this point and have altered the language around how these drugs may cooperate to reflect it (lines 255-258 and 351-353). We did not include densitometry as PIM/AURKA co-targeting is not a major conclusion of the revised study. Further, the poor detection of endogenous c-MYC with the available antibodies limit their measurement with the low-sensitivity systems used for quantification. We also acknowledge the low levels of n-MYC expression and have focused the manuscript on c-MYC given its higher level of expression and clearer role.

8. Data weak (Figure 5D): Decrease of PIM2 is not evident in Colo741 as described in text for Figure 5D. Try stimulating PKA with FSK/IMBX (as in Figure 4) in the presence and absence of siRNA to PRKACA.

This figure was intended to show a relative lack of effect in Colo741, where PIM2 appears less important, but in retrospect may have been confusing. We have removed this figure.

9. Use of AML12: This transgenic knockin of DNAJB1-PRKACA in murine hepatocytes may be the 'cleanest' cell system to interrogate the proposed PKA-AURKA/PIM-MYC axis as the only genetic alteration is the transgene (Figure 6A).

We agree that the AML12 system has many advantages. In response to the reviewer’s suggestion, we examined it further, but found that the long-term presence of the DNAJ-PKAc fusion drove parallel changes in signaling and kinase gene expression (i.e. protein levels were upregulated in parallel to phosphorylation). For this reason, genetic and pharmacological manipulation of human cells proved more illustrative and are shown throughout the manuscript.

10. Figure 6B: Many mechanisms regulate MYC expression. It remains unclear whether the PKA-AURKA/PIM-MYC axis is functional in FLC as described as Figure 6B not convincing.

This figure has been revised to more accurately reflect the balance of effects on c-MYC downstream of PKA.

Reviewer #3 (Recommendations for the authors):The study by Chan, Gordan et al., utilizes state of the art global kinome profiling to map the shared signaling networks driven by diverse genetic changes resulting in PKA activation in human cancer. Among the many kinases whose activity is modulated by active PKA, they authors centered their study on Aurora Kinase A (AURKA), and its ability to regulate c-MYC and n-MYC protein levels. They propose an AURKA-MYC regulatory network, and a possible positive feedback loop mediated by the kinase PIM2, which can be disrupted by AURKA inhibition. The study has many elements of novelty, which could be of translational importance. The strength of the study is the use of global kinome profiling and the identification of multiple candidate signaling nodes downstream from PKA. The weakness is the limited mechanistic information provided on possible direct regulatory processes intervening in kinase activation, and the need to enhance the rigor of the studies and to provide quantitative analysis of the data to increase the confidence regarding the proposed novel mechanisms at this stage.What is the mechanism by which PKA activates AURKA? Is AURKA phosphorylated by PKA? AURKA expression levels and enzymatic activity changes occur with the cell cycle, so it is unclear what the direct link between PKA and AURKA is, or if this (and other) effects are due to the impact of PKA on general cell growth or other indirect processes.Similarly, the proposed link between PKA-AURKA, MYC, and the positive feedback through PIM2 is not mechanistically defined.

We appreciate the reviewer’s positive assessment of the novelty and potential impact of our study. We have added additional data showing phosphorylation changes in proposed PKA targets including GSK3B and AURKA, noting that AURKA T288 has been previously described to be a PKA target (Walter et al., Oncogene 2000). We note that we were unable to consistently detect AURKA in our cells, and was more sensitive to nocodazole synchronization, supporting our investigation of other mechanisms. We further add phosphoproteomic data demonstrating increased phosphorylation of sets of proteins involved in translation initiation and western blot data showing increased phosphorylation of eIF4B at Ser422. These and other data fill in the missing mechanistic links from the initial submission.

Figure 1:The frequency (not the incidence?) of PRKACA amplification described in the text (0.3-3.2%) does not match with the information in the corresponding figure (1B).

This section was not clearly written and has been corrected. We apologize for any confusion.

With all the emphasis on the validity of FLX1 cells as a model for PKA-driven FLC, it is not clear why these cell were not used for proteomics analysis (Panel C).

Previous work has shown biochemical gain of function in FLX1 cells due to the DNAJ-PKAc fusion, making them different from lines with *PRKACA* amplification or *PRKAR1A* inactivation (see Turnham et al., 2019). Furthermore, these cells were not available when our initial data set was generated and have not proven as amenable to engineering. The text has been clarified to explain this.

Unless mistaken, Colo741 skin cancer cells have a frameshift mutation in PRKAR1, but the expression levels of this protein did not change (1G, third and fourth lane).

We are grateful for this sharp observation. There appears to have been an error in the initial western blot, potentially due incomplete stripping of a previously blotted protein. We re-confirmed the identity of the engineered cells using STR analysis and repeated the western blot with the cell stocks frozen at the time of our proteomic analysis. New data are shown.

There seems to be little expression of PKA C subunit in ML1 cells after dox induction.

This may also have been due to an error when the figure was prepared. The repeat blot shown in Figure 1D has a higher level and clearly demonstrates increase phosphorylation of PKA substrates.

Figure 2:The kinases whose activity is regulated by PKA C expression may be direct or indirect, and hence caution may need to be taken regarding the direct regulatory role of PKA on these kinases. For example, the authors mention that many of these kinases are involved in G2M, and hence it is possible that PKA C regulates these kinases indirectly through promoting cell cycle progression, among other indirect mechanisms.

We appreciate this point and have modified the text to note that both direct and indirect mechanisms are anticipated to be shown in these findings. We also include additional cell cycle analysis in supplementary data and Author response images 2 and 3, supporting a signaling effect of PKA on these targets.

In Sup 1, ML1 cells that are not sensitive to PKA inhibition were used, without a clear explanation. That said, cell proliferation inhibition by MEK1/2 and ERK1/2 inhibitors were less sensitive in these cells, and mentioned as probably of no interest, whereas there was a very limited increase in the sensitivity to one AURKAi, CD532, and changes in the responses to another MLN8237, which itself was not very effective, but the latter deemed of importance. This reviewer is unclear about the rationale and biological interpretation of the data. The comment on "older generation" AURKAi (ENMD-2076) affecting PKA C induction appears to be odd.

We appreciate these observations. The data from ML1 have been removed, and more comprehensive data from FLX1 substituted in Figures 5A-B. To expand on our rationale in the experiments with Aurora Kinase inhibitors, our focus has been to identify agents that are potent or whose activity is significantly modified by PKA signaling activation. As many pharmacological studies pointed us towards AURKA as a target, we investigated a range of compounds targeting Aurora kinases and noted CD532 to be particularly active. CD532 alters AURKA conformation, causing c-MYC and n-MYC degradation. However, our western blots showed that CD532 also inhibits PKA directly. Thus, we selected MLN8237 as it had been demonstrated to also alter AURKA conformation in a way that influences c-MYC expression. We have expanded on our explanation of the rationale for selecting these agents in the text.

Figure 3:Seems that only CD532 is effective in reducing proliferation of PKA C dependent cells among all AURKAi tested, raising the possibility of an off target effect.

We agree. This point has been made clearer in the text.

In that venue, changes in gene expression reported, most of which are involved in cell cycle progression, may be due to decrease cell proliferation and not acting directly downstream from PKA C, including c-MYC.

These data have been removed with more relevant data substituted (Figure 4A-C). We do note that many of the expression targets identified are still involved in cell cycle progression, but the controls reported in Author response image 1 and Figure 5, Figure 5—figure supplement 1B support a separable PKA signaling effect on c-MYC.

Data on C-MYC siRNA effects (and in fact all siRNA studies, including MYCN) require more than one siRNA and quantitative analysis of the impact on protein expression and changes.

4 individual siRNA were pooled for these experiments; we regret that this was not clearer in the initial manuscript. We have also added in significant data using individual siRNA (Figure 4C; Figure 3—figure supplement 1; Figure 7F; Figure 7—figure supplement 1C). The design of these experiments did not enable quantitative western blot analysis, but quantitative PCR is included to demonstrate the degree of knockdown. We also acknowledge that the low level expression of n-MYC made interpretation of its knockdown studies difficult. The revised manuscript has been focused specifically on c-MYC to avoid any uncertainties posed by weak detection of n-MYC by western blot and quantitative PCR.

This also applies to cell proliferation. For example, in Colo741 cells there appears to be less than 20% decrease in cell proliferation after MYC siRNA use, which is of questionable biological relevance.

We agree, this point has been made clearer in the text.

Information of cell groups (grey/red/yellow) in FLX1 experiments is missing and cannot be reviewed.

We apologize for this error; the missing label has been added.

The text on c-MYC and n-MYC knock down in Colo741 cells seems to be incorrect as these cells do not express n-MYC. The impact of n-MYC siRNA on the protein expression of n-MYC (Figure 3D) is not clear, and needs quantification as for most other similar knockdown studies.

We apologize for any lack of clarity in our description of the results. We also acknowledge that the very low levels of n-MYC make the impact of knockdown difficult to assess. Similarly, the very low levels of *MYCN* mRNA made the follow up experiments with individual siRNA difficult to assess. Accordingly, we have removed the experiments with *MYCN* knockdown and only show those with *MYC* knockdown.

Figure 4:Panel B. As for other similar studies, need to use of more than 1 siRNA for PRKACA to decrease the possibility of off-target effects.

4 individual siRNA were pooled for these experiments. We undertook experiments with individual siRNAs as well but found that a less than 50% reduction in PKA protein expression rendering it difficult to see reductions in PKA signaling. In place of this experiment, we have substituted treatment with the PKA inhibiting tool compound H89 (Figure 3E). We believe that our pooled siRNA experiments showing signaling differences are still relevant to the manuscript and do replicate the findings of both PRKAR1A^G325D^ induction and PKA inhibition with H89. This, we believe they merit inclusion as additional supportive data, but we have added further acknowledgement of their limitations.

Figure 5.Panel B. Use more than one siRNA for PIM1 and PIM2. This is emphasized by the fact that PIM2 silencing abrogates the expression of both PIM proteins.

We acknowledge that these data were lacking; they have been removed.

The use of MLN8237 and CX6258 does not seem to provide "cooperating" or even additive effects with respect to c-MYC and n-MYC expression.

We have re-worded the description of these results.

The positive feedback loop mediated by PIM2 downstream AURKA and MYC appears to be speculative, and not mechanistically explained, at the protein interaction, activity changes, and/or expression level.

We acknowledge that our prior data implicated but did not confirm this mechanism. We have removed it from the manuscript.

Figure 6:AURKA and PIM2 levels are not regulated in FLC tumor lesions harboring endogenous PKA C activating protein as proposed by the studies using ectopically expressed proteins, nor do they appear to correlate with c-MYC and n-MYC expression levels. This raises concerns about the overall cancer relevant significance of the proposed regulatory model in this study.

We acknowledge this concern and have addressed it in our restructuring of the manuscript.